# Learning excitatory-inhibitory neuronal assemblies in recurrent networks

**Owen Mackwood[1,2], Laura B Naumann[1,2], Henning Sprekeler[1,2]\***

[1]Bernstein Center for Computational Neuroscience Berlin, Berlin, Germany;
[2]Department for Electrical Engineering and Computer Science, Technische Universität Berlin, Berlin, Germany

**Abstract** Understanding the connectivity observed in the brain and how it emerges from local plasticity rules is a grand challenge in modern neuroscience. In the primary visual cortex (V1) of mice, synapses between excitatory pyramidal neurons and inhibitory parvalbumin-expressing (PV) interneurons tend to be stronger for neurons that respond to similar stimulus features, although these neurons are not topographically arranged according to their stimulus preference. The presence of such excitatory-inhibitory (E/I) neuronal assemblies indicates a stimulus-specific form of feedback inhibition. Here, we show that activity-dependent synaptic plasticity on input and output synapses of PV interneurons generates a circuit structure that is consistent with mouse V1. Computational modeling reveals that both forms of plasticity must act in synergy to form the observed E/I assemblies. Once established, these assemblies produce a stimulus-specific competition between pyramidal neurons. Our model suggests that activity-dependent plasticity can refine inhibitory circuits to actively shape cortical computations.

## Introduction

With the advent of modern optogenetics, the functional role of inhibitory interneurons has developed into one of the central topics of systems neuroscience (*Fishell and Kepecs, 2020*). Aside from the classical perspective that inhibition serves to stabilize recurrent excitatory feedback loops in neuronal circuits (*van Vreeswijk and Sompolinsky, 1996*; *Brunel, 2000*; *Murphy and Miller, 2009*; *Sprekeler, 2017*), it is increasingly recognised as an active player in cortical computation (*Isaacson and Scanziani, 2011*; *Priebe and Ferster, 2008*; *Rubin et al., 2015*; *Pouille and Scanziani, 2001*; *Letzkus et al., 2011*; *Adesnik et al., 2012*; *Hennequin et al., 2014*; *Phillips et al., 2017*; *Barron et al., 2016*; *Barron et al., 2017*; *Tovote et al., 2015*).

Within cortical neurons, excitatory and inhibitory currents are often highly correlated in their response to stimuli (*Wehr and Zador, 2003*; *Froemke et al., 2007*; *Tan et al., 2011*; *Bhatia et al., 2019*), in time (*Okun and Lampl, 2008*; *Dipoppa et al., 2018*) and across neurons (*Xue et al., 2014*). This co-tuning of excitatory and inhibitory currents has been attributed to different origins. In topographically organised sensory areas such as cat primary visual cortex (V1), the co-tuning with respect to sensory stimuli could be a natural consequence of local feedback inhibition and does not impose strong constraints on inhibitory circuitry (*Harris and Mrsic-Flogel, 2013*). In the case of feed-forward inhibition, co-tuning of excitatory and inhibitory currents was suggested to arise from homeostatic synaptic plasticity in GABAergic synapses (*Vogels et al., 2011*; *Clopath et al., 2016*; *Weber and Sprekeler, 2018*; *Hennequin et al., 2017*).

In sensory areas with poor feature topography, such as V1 of rodents (*Ohki et al., 2005*), feedback inhibition has been hypothesised to be largely unspecific for stimulus features, a property inferred from the dense connectivity (*Fino and Yuste, 2011*; *Packer and Yuste, 2011*) and reliable presence of synapses connecting pyramidal (Pyr) neurons to inhibitory interneurons with dissimilar stimulus tuning (*Harris and Mrsic-Flogel, 2013*; *Bock et al., 2011*; *Hofer et al., 2011*). However,

\*For correspondence:
h.sprekeler@tu-berlin.de

**Competing interests:** The authors declare that no competing interests exist.

recent results cast doubt on this idea of a 'blanket of inhibition' (*Fino and Yuste, 2011*; *Packer and Yuste, 2011*).

In mouse V1, *Znamenskiy et al., 2018* report that although the presence of synaptic connections between Pyr cells and parvalbumin-expressing (PV) interneurons is independent of their respective stimulus responses, the efficacy of those synapses is correlated with their response similarity, both in PV → Pyr and in Pyr → PV connections. These mutual preferences in synaptic organisation suggest that feedback inhibition may be more stimulus-specific than previously thought and that Pyr and PV neurons form specialised—albeit potentially overlapping—excitatory-inhibitory (E/I) assemblies (*Chenkov et al., 2017*; *Yoshimura et al., 2005*; *Litwin-Kumar and Doiron, 2012*; *Litwin-Kumar and Doiron, 2014*). While the presence of such E/I assemblies (*Znamenskiy et al., 2018*; *Rupprecht and Friedrich, 2018*) suggests the need for an activity-dependent mechanism for their formation and/or refinement (*Khan et al., 2018*; *Najafi et al., 2020*), the requirements such a mechanism must fulfil remain unresolved.

Here, we use a computational model to identify requirements for the development of stimulus-specific feedback inhibition. We find that the formation of E/I assemblies requires a synergistic action of plasticity on two synapse types: the excitatory synapses from Pyr neurons onto PV interneurons and the inhibitory synapses from those interneurons onto the Pyr cells. Using 'knock-out experiments', in which we block plasticity in either synapse type, we show that both must be plastic to account for the observed functional microcircuits in mouse V1. In addition, after the formation of E/I assemblies, perturbations of individual Pyr neurons lead to a feature-specific suppression of other Pyr neurons as recently found in mouse V1 (*Chettih and Harvey, 2019*). Thus, synergistic plasticity of the incoming and outgoing synapses of PV interneurons can drive the development of stimulus-specific feedback inhibition, resulting in a competition between Pyr neurons with similar stimulus preference.

## Results

To understand which activity-dependent mechanisms can generate specific feedback inhibition in circuits without feature topography—such as mouse V1 (*Figure 1a*), we studied a rate-based network model consisting of $N^{\mathrm{E}} = 512$ excitatory Pyr neurons and $N^{\mathrm{I}} = 64$ inhibitory PV neurons. To endow the excitatory neurons with a stimulus tuning similar to Pyr cells in layer 2/3 of mouse V1 (*Znamenskiy et al., 2018*), each excitatory neuron receives external excitatory input that is tuned to orientation, temporal frequency and spatial frequency (*Figure 1b*). The preferred stimuli of the Pyr neurons cover the stimulus space evenly. Because we are interested under which conditions feedback inhibition can acquire a stimulus selectivity, inhibitory neurons receive external inputs without stimulus tuning, but are recurrently connected to Pyr neurons. While the network has no stimulus topography, Pyr neurons are preferentially connected to other Pyr neurons with similar stimulus tuning (*Hofer et al., 2011*; *Cossell et al., 2015*), and connection strength is proportional to the signal correlation of their external inputs. Note that the Pyr → Pyr connections only play a decisive role for the results in Figure 4 but are present in all simulations for consistency. Connection probability across the network is $p = 0.6$, with the remaining network connectivity (Pyr → PV, PV → PV, PV → Pyr) initialised randomly according to a log-normal distribution (*Song et al., 2005*; *Loewenstein et al., 2011*), with a variability that is similar to that measured in the respective synapses (*Znamenskiy et al., 2018*).

### E/I assemblies are formed by homeostatic plasticity rules in input and output connections of PV interneurons

In feedforward networks, a stimulus-specific balance of excitation and inhibition can arise from homeostatic inhibitory synaptic plasticity that aims to minimise the deviation of a neuron's firing rate from a target for all stimuli of a given set (*Vogels et al., 2011*; *Clopath et al., 2016*; *Weber and Sprekeler, 2018*). We wondered whether a stimulus-specific form of homeostasis can also generate stimulus-specific *feedback* inhibition by forming E/I assemblies. To that end, we derive synaptic plasticity rules for excitatory input and inhibitory output connections of PV interneurons that are homeostatic for the excitatory population (see 'Materials and methods'). A stimulus-specific homeostatic control can be seen as a 'trivial' supervised learning task, in which the objective is that all Pyr neurons should learn to fire at a given target rate $\rho_0$ for all stimuli. Hence, a gradient-based optimisation

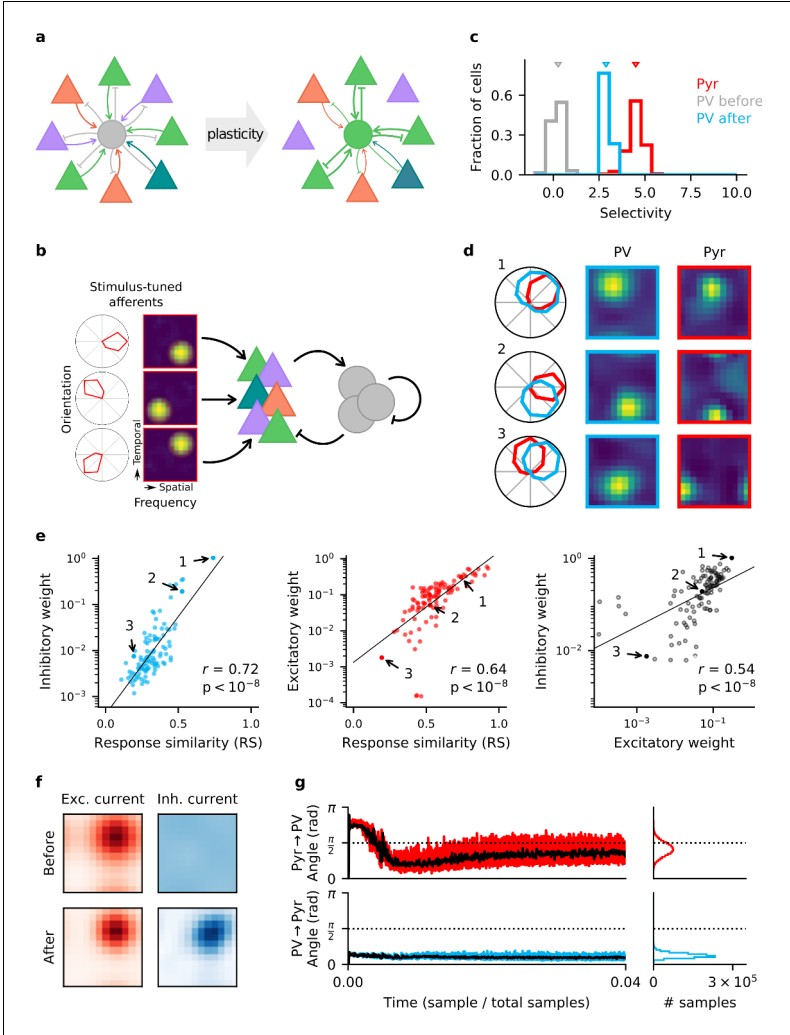

**Figure 1.** Homeostatic plasticity in input and output synapses of interneurons drives the formation of excitatory-inhibitory (E/I) assemblies. (**a**) Emergence of E/I assemblies comprised of pyramidal (Pyr) neurons (triangles) and parvalbumin-expressing (PV) interneurons (circles) in circuits without feature topography. (**b**) Network architecture and stimulus tuning of external inputs to Pyr cells. (**c**) Stimulus selectivity of Pyr neurons and PV interneurons (before and after learning). Arrows indicate the median. (**d**) Example responses of reciprocally connected Pyr cells and PV interneurons. Examples chosen for large, intermediate, and low response similarity (RS). Numbers correspond to points marked in (**e**). (**e**) Relationship of synaptic efficacies of output (left) and input connections (centre) of PV interneurons with RS. Relationship of input and output efficacies (right). Black lines are obtained via linear regression. Reported $r$ and associated p-value are Pearson's correlation. (**f**) Stimulus tuning of excitatory and inhibitory currents onto an example Pyr cell, before and after learning. For simplicity, currents are shown for spatial and temporal frequency only, averaged across all orientations. (**g**) Angle between the weight update and the gradient rule while following the local approximation for input (top) and output (bottom) connections of PV interneurons. Time course for first 4% of simulation (left) and final distribution (right) shown. Black lines are low-pass filtered time courses.

The online version of this article includes the following figure supplement(s) for figure 1:

**Figure supplement 1.** Synaptic plasticity and convergence.

**Figure supplement 2.** Gradient rules also require plasticity of both input and output synapses of parvalbumin-expressing (PV) interneurons.

**Figure supplement 3.** Synaptic currents onto pyramidal (Pyr) neurons.

**Figure supplement 4.** Both input and output synapses must be plastic for feedback alignment to occur.

**Figure supplement 5.** Some networks contain experimentally undetectable weights.

would effectively require a backpropagation of error (*Rumelhart et al., 1985*) through time (BPTT; *Werbos, 1990*).

Because backpropagation rules rely on non-local information that might not be available to the respective synapses, their biological plausibility is currently debated (*Lillicrap et al., 2020*; *Sacramento et al., 2018*; *Guerguiev et al., 2017*; *Whittington and Bogacz, 2019*; *Bellec et al., 2020*). However, a local approximation of the full BPTT update can be obtained under the following assumptions: First, we assume that the sensory input to the network changes on a time scale that is slower than the intrinsic time scales in the network. This eliminates the necessity of backpropagating information through time, albeit still through the synapses in the network. This assumption results in what we call the 'gradient-based' rules (*Equation 15* in Appendix 1), which are spatially non-local. Second, we assume that synaptic interactions in the network are sufficiently weak that higher-order synaptic interactions can be neglected. Third and finally, we assume that over the course of learning, the Pyr → PV connections and the PV → Pyr connections become positively correlated (*Znamenskiy et al., 2018*), such that we can replace PV → Pyr synapses by the reciprocal Pyr → PV synapse in the Pyr → PV learning rule, without rotating the update too far from the true gradient (see Appendix 1).

The resulting learning rule for the output connections of the interneurons is similar to a previously suggested form of homeostatic inhibitory plasticity (*Figure 1—figure supplement 1a*, left) (*Vogels et al., 2011*). Specifically, PV output synapses $W^{E \leftarrow I}$ undergo Hebbian changes in proportion to presynaptic interneuron activity $r_j^I$ and the signed deviation of total postsynaptic Pyr cell input $h_i^E$ from the homeostatic target:

$$\Delta W_{ij}^{E \leftarrow I} \propto r_j^I (h_i^E - \rho_0) + \text{weight decay}.$$

In contrast, the PV input synapses $W^{I \leftarrow E}$ are changed such that the total excitatory drive $I_i^{E,rec}$ from the Pyr population to each interneuron is close to some target value $I_0$ (*Figure 1—figure supplement 1a*, right):

$$\Delta W_{ij}^{I \leftarrow E} \propto r_j^E (I_i^{E,rec} - I_0) + \text{weight decay}.$$

Both synapse types are subject to a weak weight decay, to avoid the redundancy that a multiplicative rescaling of input synapses can be compensated by a rescaling of the output synapses.

While our main results are obtained using the local approximations, we also simulated the gradient-based rules to verify that the approximation does not qualitatively change the results (*Figure 1—figure supplement 2*).

When we endow the synapses of an initially randomly connected network of Pyr neurons and PV interneurons with plasticity in both the input and the output synapses of the interneurons, the network develops a synaptic weight structure and stimulus response that closely resemble that of mouse V1 (*Znamenskiy et al., 2018*). Before learning, interneurons show poor stimulus selectivity (*Figure 1c*), in line with the notion that in a random network, interneurons pool over many Pyr neurons with different stimulus tuning (*Harris and Mrsic-Flogel, 2013*). The network is then exposed to randomly interleaved stimuli. By the end of learning, interneurons have developed a pronounced stimulus tuning, albeit weaker than that of Pyr neurons (*Figure 1c,d*). Interneurons form strong bidirectional connections preferentially with Pyr neurons with a similar stimulus tuning, whereas connections between Pyr-PV pairs with dissimilar stimulus tuning are weaker (*Figure 1d,e*). To make our results comparable to *Znamenskiy et al., 2018*, we randomly sample an experimentally feasible number of synaptic connections from the network ($n = 100$). Both the efficacy of PV input and output connections are highly correlated with the response similarity (RS) (see 'Materials and methods') of the associated Pyr neurons and interneurons (*Figure 1e*, left and centre). For bidirectionally connected cell pairs, the efficacies of the respective input and output connections are highly correlated (*Figure 1e*, right). The stimulus tuning of the inhibitory inputs onto the Pyr cells—initially flat—closely resembles that of the excitatory inputs after learning (*Figure 1f*, *Figure 1—figure supplement 3*; *Tan et al., 2011*), that is, the network develops a precise E/I balance (*Hennequin et al., 2017*).

Finally, the optimal gradient rules produce very similar results to the local approximations (*Figure 1—figure supplement 2*). Over the course of learning, the weight updates by the approximate

rules align to the updates that would result from the gradient rules (*Figure 1g*, *Figure 1—figure supplement 4*), presumably by a mechanism akin to feedback alignment (*Lillicrap et al., 2016*; *Akrout et al., 2019*).

In summary, these results show that combined homeostatic plasticity in input and output synapses of interneurons can generate a similar synaptic structure as observed in mouse V1, including the formation of E/I assemblies.

## PV → Pyr plasticity is required for the formation of E/I assemblies

Having shown that homeostatic plasticity acting on both input and output synapses of interneurons are *sufficient* to learn E/I assemblies, we now turn to the question of whether both are *necessary*. To this end, we perform 'knock-out' experiments, in which we selectively block synaptic plasticity in either of the synapses. The motivation for these experiments is the observation that the incoming PV synapses follow a long-tailed distribution (*Znamenskiy et al., 2018*). This could provide a sufficient stimulus selectivity in the PV population for PV → Pyr plasticity alone to achieve a satisfactory E/I balance. A similar reasoning holds for static, but long-tailed outgoing PV synapses. This intuition is supported by results from *Litwin-Kumar et al., 2017*, where for a population of neurons analogous to our interneurons, the dimensionality of responses in that population can be high for static input synapses, when those are log-normally distributed.

When we knock out output plasticity but keep input plasticity intact, the network fails to develop E/I assemblies and a stimulus-specific E/I balance. While there is highly significant change in the distribution of PV interneuron stimulus selectivity (Mann-Whitney $U$ test, $U = 1207$, $p<10^{-4}$), the effect is much stronger when output plasticity is also present (*Figure 2a,b*). Importantly, excitatory and inhibitory currents in Pyr neurons are poorly co-tuned (*Figure 2c*, *Figure 1—figure supplement 3b*). In particular, feedback inhibition remains largely untuned because output connections are still random, so that Pyr neurons pool inhibition from many interneurons with different stimulus tuning.

To investigate whether the model without output plasticity is consistent with the synaptic structure of mouse V1, we repeatedly sample an experimentally feasible number of synapses ($n = 100$, *Figure 2d*) and plot the distribution of the three pairwise Pearson's correlation coefficients between the two classes of synaptic weights and RS (*Figure 2e*). When both forms of plasticity are present in the network, a highly significant positive correlation ($p<0.01$) is detected in all samples for all three correlation types (*Figure 2f*). When output plasticity is knocked out, we still find a highly significant positive correlation between input weights and RS in 99% of the samples (*Figure 2d–f*). In contrast, correlations between input and output synapses are weaker and cannot reliably be detected (2% of samples). Notably, we find a correlation between output weights and RS in <0.01% of samples (*Figure 2f*). Finally, for an experimentally realistic sample size of $n = 100$, the probability of a correlation coefficient equal or higher than that observed by *Znamenskiy et al., 2018* is <0.01% for the correlation between output weights and RS ($r = 0.55$), and <0.01% for the correlation between input and output synapses ($r = 0.52$).

The non-local gradient rule for the PV input synapses alone also does not permit the formation of E/I assemblies (*Figure 1—figure supplement 2*). While the selectivity of interneurons increases more than for the local approximation (*Figure 1—figure supplement 2b*), feedback inhibition still remains untuned in the absence of output plasticity (*Figure 1—figure supplement 2c,d*).

We therefore conclude that input plasticity alone is insufficient to generate the synaptic microstructure observed in mouse V1.

## Pyr → PV plasticity is required for assembly formation

When we knock out input plasticity but keep output plasticity intact, we again observe no formation of E/I assemblies. This remains true even when using the gradient-based rule (*Figure 1—figure supplement 2*). The underlying reason is that input weights remain random. Interneurons collect excitation from many Pyr neurons with different preferences, and absent plasticity on their input synapses, they maintain their initial poor stimulus selectivity (*Figure 3a–c*). Because of the poor stimulus tuning of the interneurons, output plasticity cannot generate stimulus-specific inhibitory inputs to the Pyr neurons (*Figure 3d*). Instead, they essentially receive a tonic, unspecific background inhibition that is weakly modulated by the stimulus (*Figure 1—figure supplement 3b*). While this weak modulation is

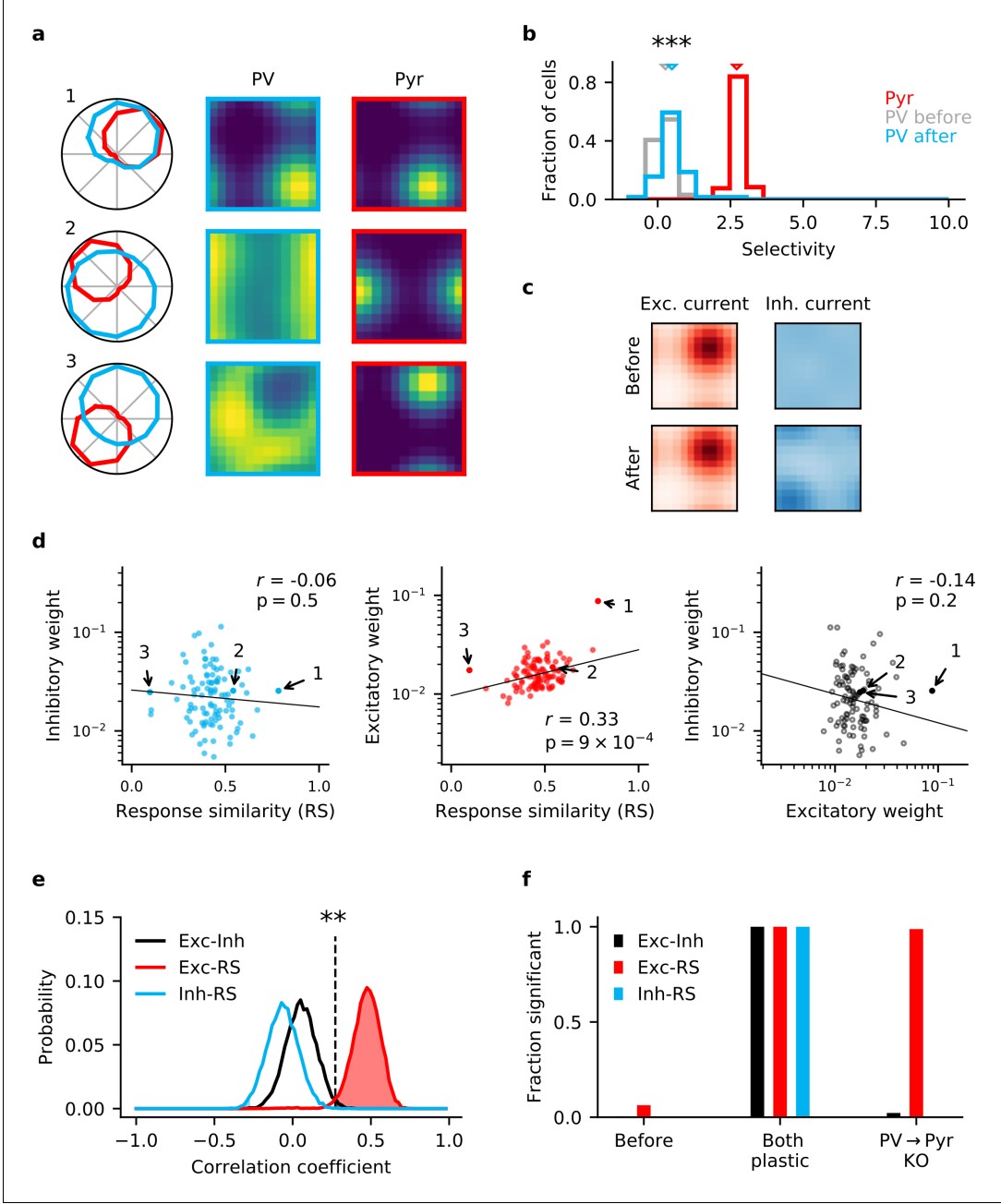

**Figure 2.** Knock-out (KO) of plasticity in parvalbumin-expressing (PV) interneuron output connections prevents inhibitory co-tuning. (**a**) Example responses of reciprocally connected pyramidal (Pyr) cells and PV interneurons. Numbers correspond to points marked in (**d**). (**b**) Stimulus selectivity of Pyr cells and PV interneurons (before and after learning; Mann-Whitney $U$ test, $p<10^{-4}$). Arrows indicate median. (**c**) Stimulus tuning of excitatory and inhibitory input currents in a Pyr cell before and after learning. For simplicity, currents are shown for spatial and temporal frequency only, averaged across all orientations. (**d**) Relationship of output (left) and input (centre) synaptic efficacies of PV interneurons with response similarity. Relationship of input and output efficacies (right). Plotted lines are obtained via linear regression. Reported $r$ and associated p-value are the Pearson's correlation. (**e**) Distribution of Pearson's correlation coefficients for multiple samples as shown in (**d**). Dashed line marks threshold of high significance ($p<0.01$). (**f**) Fraction of samples with highly significant positive correlation before plasticity, after plasticity in both input and output connections, and for KO of plasticity in PV output connections (based on 10,000 random samples of 100 synaptic connections).

correlated with the excitatory inputs, the overall similarity between excitatory and inhibitory input remains low (*Figure 1—figure supplement 3c*).

This modulation is made possible by the fact that interneurons still possess a weak, but consistent stimulus tuning arising from random variations in their input weights. A particularly strong input connection will cause the postsynaptic interneuron to prefer similar stimuli to the presynaptic Pyr. Because of the resulting correlated activity, the Hebbian nature of the output plasticity potentiates inhibitory weights for such cell pairs that are reciprocally connected. The tendency of strong input

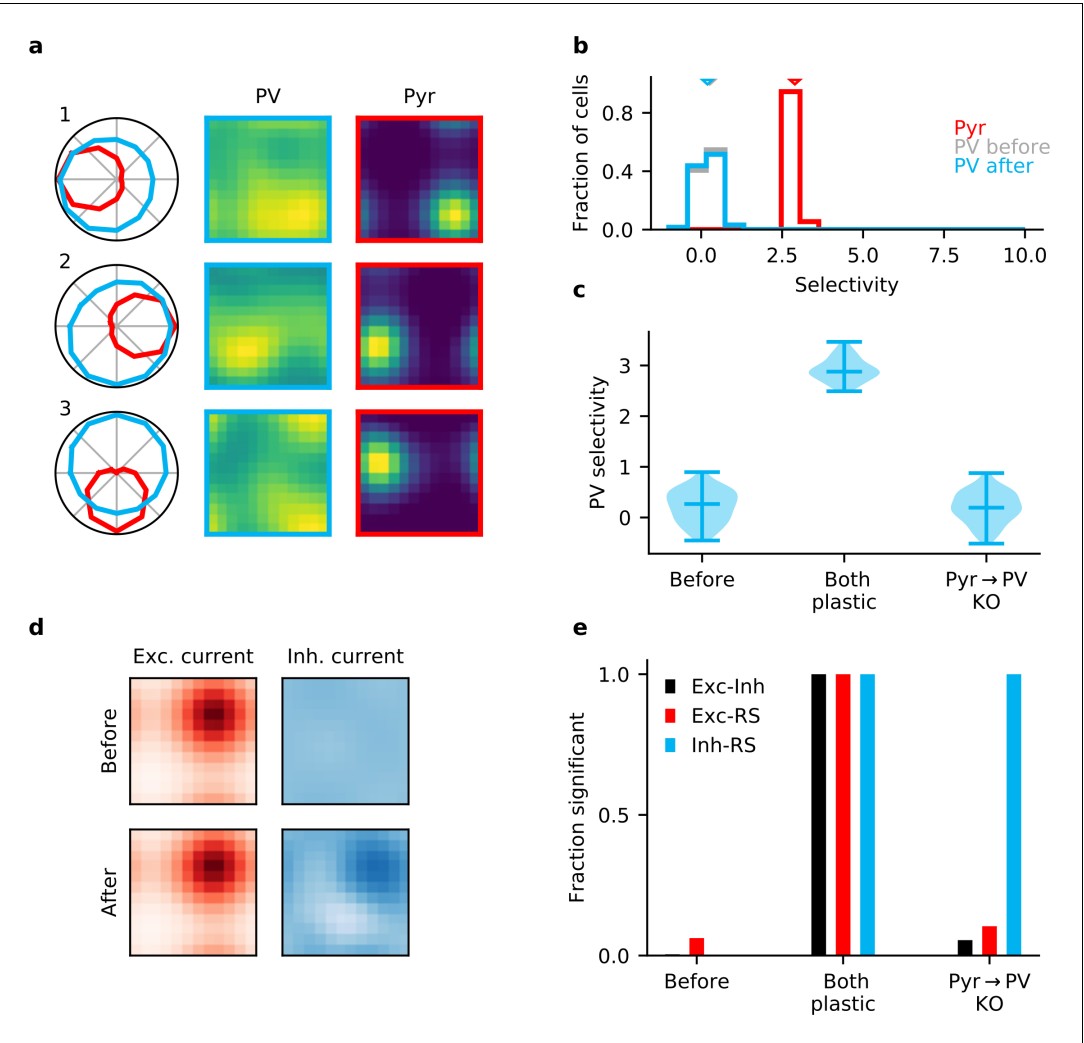

**Figure 3.** Plasticity of parvalbumin-expressing (PV) interneuron input connections is required for inhibitory stimulus selectivity and current co-tuning. (a) Example responses of reciprocally connected pyramidal (Pyr) cells and PV interneurons. (b) Stimulus selectivity of Pyr cells and PV interneurons (before and after learning). Arrows indicate median. (c) Violin plots of inhibitory stimulus selectivity before plasticity, after learning with plasticity in both input and output connections of PV interneurons and for knock-out (KO) of plasticity in PV input connections. (d) Stimulus tuning of excitatory and inhibitory currents in a Pyr cell before and after learning. Dimensions correspond to spatial and temporal frequency of the stimuli averaged across all orientations. (e) Fraction of samples with highly significant (*p*<0.01) positive correlation before plasticity, after plasticity in both input and output connections, and for KO of plasticity in PV input connections (based on 10,000 random samples of 100 synaptic connections).

The online version of this article includes the following figure supplement(s) for figure 3:

**Figure supplement 1.** Correlation between weights and response similarity.

**Figure supplement 2.** Long-tailed Pyr → PV weight distribution does not reproduce experimentally observed correlations.

synapses to generate a strong corresponding output synapse is reflected in a positive correlation between output synapses and RS (*Figure 3e*, *Figure 3—figure supplement 1*), despite the fact that input synapses remain random.

This effect further increases when input synapses are drawn from a distribution with an even heavier tail, beyond what is observed in mouse V1 (*Znamenskiy et al., 2018*; *Figure 3—figure supplement 2a*). In this case, the stimulus tuning of the interneurons is dominated by a small number of very large synapses. The resulting higher selectivity of the interneurons (*Figure 3—figure supplement 2b*) allows a better co-tuning of excitation and inhibition in Pyr neurons (*Figure 3—figure supplement 2c*), in line with theoretical arguments for sparse connectivity (*Litwin-Kumar et al., 2017*). However, the dominance of a small number of large synapses also makes it unlikely that those synapses are observed in an experiment in which a finite number of synapses are sampled. As a result, a heavier tail does not yield the correlation of reciprocal input and output synapses observed by *Znamenskiy et al., 2018* (*Figure 3—figure supplement 2d,e*), although it increases the probability of observing correlations between input synapses and RS when weak synapses are discarded. See Appendix 1 for a more extensive discussion.

Collectively, these results indicate that plasticity of both the inhibitory output and the excitatory input synapses of PV interneurons is required for the formation of E/I assemblies in cortical areas without feature topography, such as mouse V1.

## Single-neuron perturbations

Our findings demonstrate that in networks without feature topography, only a synergy of excitatory and inhibitory plasticity can account for the emergence of E/I assemblies. But how does stimulus-specific feedback inhibition affect interactions between excitatory neurons? In layer 2/3 of V1, similarly tuned excitatory neurons tend to have stronger and more frequent excitatory connections (*Ko et al., 2011*). It has been hypothesised that this tuned excitatory connectivity supports reliable stimulus responses by amplifying the activity of similarly tuned neurons (*Cossell et al., 2015*). However, the presence of co-tuned feedback inhibition could also induce the opposite effect, such that similarly tuned excitatory neurons are in competition with each other (*Chettih and Harvey, 2019*; *Moreno-Bote and Drugowitsch, 2015*).

To investigate the effect of stimulus-specific inhibition in our network, we simulate the perturbation experiment of *Chettih and Harvey, 2019*: First, we again expose the network to the stimulus set, with PV input and output plasticity in place to learn E/I assemblies. Second, both before and after learning, we probe the network with randomly selected stimuli from the same stimulus set, while perturbing a single Pyr cell with additional excitatory input, and measure the resulting change in activity of other Pyr neurons in the network (*Figure 4a*).

While the activity of the perturbed neuron increases, many of the other Pyr neurons are inhibited in response to the perturbation (*Figure 4b*). Although comparing the pairwise influence of Pyr neurons on each other does not reveal any apparent trend (*Figure 4c*), recent experiments report that the influence a single-cell perturbation has on other neurons depends on the similarity of their stimulus feature tuning (*Chettih and Harvey, 2019*). To test whether we observe the same feature-specific suppression, we compute the influence of perturbing a Pyr on the rest of the network as a function of the receptive field correlation of the perturbed cell and each measured cell. In line with recent perturbation studies (*Chettih and Harvey, 2019*; *Sadeh and Clopath, 2020*), we observe that—on average—neurons are more strongly inhibited if they have a similar tuning to the perturbed neuron (*Figure 4d*). The opposite holds before learning: the effect of single-neuron perturbations on the network is increasingly excitatory as receptive field correlation increases. Notably, the networks in which input or output plasticity was knocked out during learning (and therefore did not develop E/I assemblies) show the same excitatory effect (*Figure 4d*, *Figure 4—figure supplement 1b*). This confirms that a 'blanket of inhibition' does not account for feature-specific suppression between excitatory neurons (*Sadeh and Clopath, 2020*).

To better understand this behaviour, we use the Pyr-Pyr receptive field correlations to compute the coefficient of determination for all pairs ($R^2$, which quantifies how well the receptive field of one Pyr neuron predicts that of another). Learning changes the correlative structure in the network (*Figure 4—figure supplement 1a*) and thereby decreases the coefficient of determination on average, indicating a reduction in Pyr-Pyr correlations within the network ($\mathrm{E}[R^2] = 0.06$ before learning, 0.02

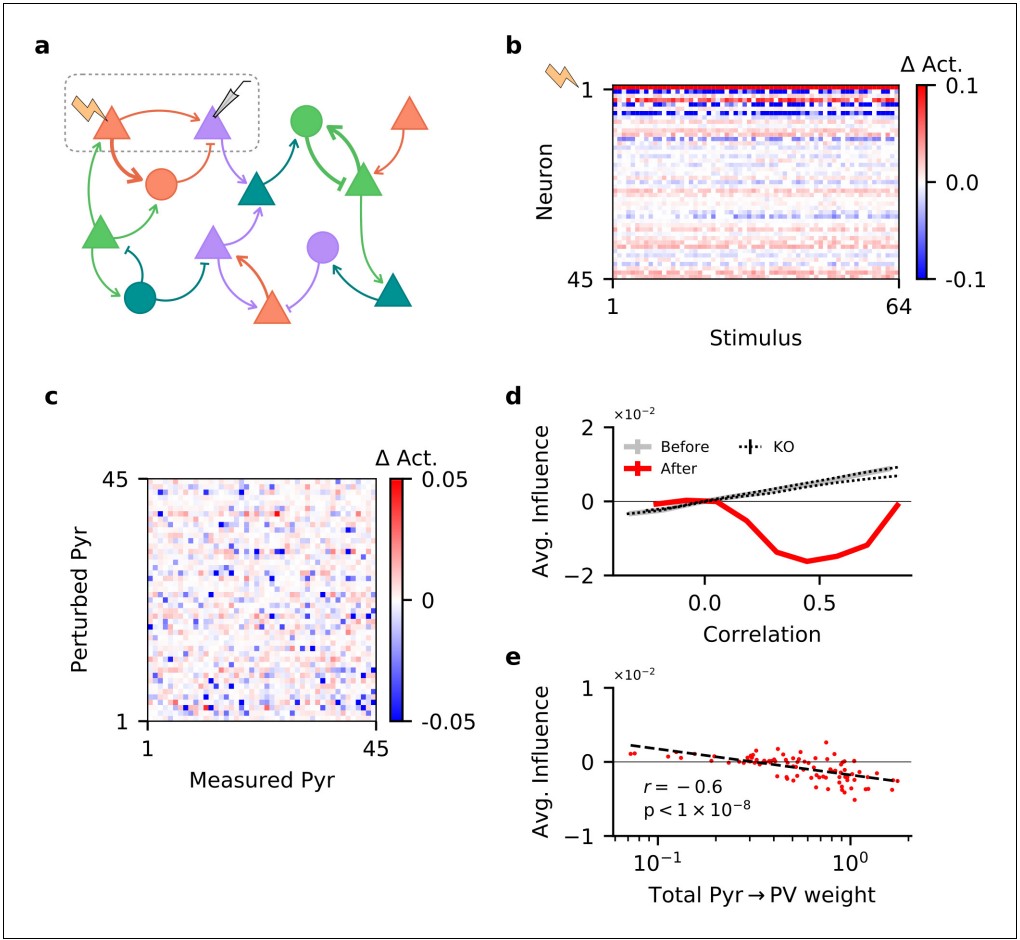

**Figure 4.** Single-neuron perturbations suppress responses of similarly tuned neurons. (**a**) Perturbation of a single pyramidal (Pyr) neuron. Responses of other Pyr neurons are recorded for different stimuli, both with and without perturbation. (**b**) Perturbation-induced change in activity (Δ Act.) of a subset of Pyr cells, for a random subset of stimuli (with neuron 1 being perturbed). (**c**) Influence of perturbing a Pyr neuron on the other Pyr neurons, averaged across all stimuli, for a subset of Pyr neurons. (**d**) Dependence of influence among Pyr neurons on their receptive field correlation (Pearson's *r*), across all neurons in the network (see 'Materials and methods'). Dotted lines indicate plasticity knock-out (KO) experiments; see *Figure 4—figure supplement 1b* for details. Error bars correspond to the standard error of the sample mean, but are not visible due to their small values. (**e**) Total strength of output synapses from a Pyr neuron predicts the average effect perturbing it has on other neurons. Dashed line is the result of a linear regression, while *r* and its associated p-value correspond to the Pearson's correlation.

The online version of this article includes the following figure supplement(s) for figure 4:

**Figure supplement 1.** Input and output plasticity together change correlations between pyramidal (Pyr) neurons, while plasticity knock-out (KO) eliminates feature competition.

---

after). Thus, plasticity suppresses some of the strongest correlations, resulting in 'feature competition' which is believed to aid sensory processing (*Lochmann et al., 2012*; *Moreno-Bote and Drugowitsch, 2015*).

While on average the network exhibits feature competition, the influence of individual Pyr neurons on the rest of the network is highly variable. According to recent modelling work (*Sadeh and Clopath, 2020*), the strength of Pyr → PV synapses strongly influences whether a network will exhibit feature competition. In our network, the total outgoing weight of a Pyr cell onto the PV neurons indeed predicts the average influence that neuron will have on the rest of the network when perturbed (*Figure 4e*; $r = -0.6$).

In summary, the stimulus-specific feedback inhibition that emerges in the model also captures the paradoxical suppression of similarly tuned excitatory neurons observed in single-cell perturbation experiments.

## Discussion

The idea that feedback inhibition serves as a 'blanket of inhibition' (*Packer and Yuste, 2011*; *Fino and Yuste, 2011*) that can be selectively broken (*Karnani et al., 2016*) has been gradually relaxed over recent years and replaced by the notion that feedback inhibition can be rather selective (*Rupprecht and Friedrich, 2018*) and could thereby support specific neuronal computations (*Vogels and Abbott, 2009*; *Hennequin et al., 2014*; *Denève and Machens, 2016*; *Najafi et al., 2020*), even in networks without topographic organisation (*Znamenskiy et al., 2018*; *Rupprecht and Friedrich, 2018*). Here, we used a computational model to show that the development of E/I assemblies similar to those observed in mouse V1 (*Znamenskiy et al., 2018*) or zebrafish olfactory areas (*Rupprecht and Friedrich, 2018*) can be driven by a homeostatic form of plasticity of the incoming and outgoing synapses of inhibitory interneurons. Based on the results of virtual knock-out experiments, we suggest that, on their own, input or output plasticity of interneurons are insufficient to explain the Pyr-PV microcircuitry in mouse V1 and that input and output plasticity in interneurons must act in synergy for stimulus-specific feedback inhibition to develop. To investigate how the presence of E/I assemblies affects interactions between excitatory neurons, we mimicked a perturbation experiment and found that—as in mouse visual cortex—stimulating single excitatory cells paradoxically suppresses similarly tuned neurons (*Chettih and Harvey, 2019*). Our findings suggest that, by driving the development of tuned feedback inhibition, plasticity of interneurons can fundamentally shape cortical processing.

The learning rules for the input and output synapses of PV interneurons are based on a single homeostatic objective that aims to keep the net synaptic current onto Pyr neurons close to a given target for all stimuli. The two forms of plasticity fulfil different purposes, however. Plasticity of input synapses is required for interneurons to acquire a stimulus selectivity, whereas plasticity of output synapses can exploit interneuron selectivity to shape inhibitory currents onto excitatory cells. The output plasticity we derived for our recurrent network is very similar to a previously suggested form of inhibitory plasticity (*Vogels et al., 2011*; *Sprekeler, 2017*). Homeostatic plasticity rules for inhibitory synapses are now used regularly in computational studies to stabilise model circuits (*Vogels et al., 2011*; *Hennequin et al., 2017*; *Landau et al., 2016*). In contrast, a theoretically grounded approach for the plasticity of excitatory input synapses onto inhibitory neurons is missing.

Homeostatic changes in excitatory synapses onto interneurons in response to lesions or sensory deprivation have been reported (*Keck et al., 2011*; *Takesian et al., 2013*; *Kuhlman et al., 2013*), but the specific mechanisms and functions of this form of interneuron plasticity are not resolved. The plasticity rule we derived for the input synapses of interneurons effectively changes the selectivity of those neurons according to the demands of the Pyr cells, that is, such that the interneurons can best counteract deviations of Pyr activity from the target. By which mechanisms such a (nearly teleological) form of plasticity can be achieved is at its core a problem of credit assignment, whose biological implementation remains open (*Lillicrap et al., 2016*; *Guerguiev et al., 2017*; *Sacramento et al., 2018*).

Here, we used a local approximation of the gradient, backpropagation rules, which produces qualitatively similar results, and which we interpret as a recurrent variant of feedback alignment, applied to the specific task of a stimulus-specific E/I balance (*Lillicrap et al., 2016*; *Akrout et al., 2019*). The excitatory input connections onto the interneurons serve as a proxy for the transpose of the output connections. The intuition why this replacement is reasonable is the following: The task of balancing excitation by feedback inhibition favours symmetric connections, because excitatory cells that strongly drive a particular PV interneuron should receive a strong feedback connection in return. Therefore, E/I balance favours a positive correlation between the incoming and outgoing synapses of PV neurons and thus the two weight matrices will be aligned in a final balanced state (*Lillicrap et al., 2016*; *Akrout et al., 2019*). This weight replacement effectively replaces the 'true' feedback errors by a deviation of the total excitatory input to the PV neurons from a target (*Hertäg and Sprekeler, 2020*). The rule therefore has the structure of a homeostatic rule for the recurrent excitatory drive received by PV neurons.

A cellular implementation of such a plasticity rule would require the following ingredients: (i) a signal that reflects the cell-wide excitatory current and (ii) a mechanism that changes Pyr → PV synapses in response to variations in this signal. For the detection of excitatory inputs, postsynaptic sodium or calcium concentrations are natural candidates. Due to the lack of spines in PV dendrites, both are expected to diffuse more broadly in the dendritic arbor than in spiny neurons (*Hu et al., 2014*; *Kullmann and Lamsa, 2007*) and may thus provide a signal for overall dendritic excitatory currents. Depending on how excitatory inputs are distributed on PV interneuron dendrites (*Larkum and Nevian, 2008*; *Jia et al., 2010*; *Grienberger et al., 2015*), the integration of the excitatory currents may not need to be cell-wide—which could limit the temporal resolution of the plasticity—but could be local, for example, to a dendrite, if local excitatory input is a sufficient proxy for the global input. Notably, in PV interneurons, NMDA receptors are enriched in excitatory feedback relative to feedforward connections (*Le Roux et al., 2013*), suggesting those two sources of excitation are differentially treated on the postsynaptic side. As for many other excitatory synapses (*Sjöström et al., 2008*), postsynaptic calcium is likely a key factor also for the plasticity of excitatory input synapses onto interneurons. Blocking NMDA receptors interferes with Hebbian long-term plasticity in some of these synapses (*Lamsa et al., 2007*; *Kullmann and Lamsa, 2007*), as does a block of excitatory input (*Le Roux et al., 2013*). Furthermore, NMDAR-dependent plasticity in Pyr → PV synapses is expressed postsynaptically and seems to require presynaptic activation (*Kullmann and Lamsa, 2007*). In summary, we believe that there are no conceptual issues that would rule out an implementation of the suggested plasticity rule for excitatory inputs onto PV interneurons.

We also expect that the rules we suggest here are only one set of many that can establish E/I assemblies. Given that the role of the input plasticity in the interneurons is the formation of a stimulus specificity, it is tempting to assume that this could equally well be achieved by classical forms of plasticity like the Bienenstock-Cooper-Munro (BCM) rule (*Bienenstock et al., 1982*), which is commonly used in models of receptive field formation. However, in our hands, the combination of BCM plasticity in Pyr → PV synapses with homeostatic inhibitory plasticity in the PV → Pyr synapses showed complex dynamics, an analysis of which is beyond the scope of this article. In particular, this combination of rules often did not converge to a steady state, probably for the following reason. BCM rules tend to make the postsynaptic neuron as stimulus-selective as possible. Given the limited number of interneurons in our circuit, this can lead to a situation in which parts of stimulus space are not represented by any interneurons. As a result, Pyr neurons that respond to those stimuli cannot recruit inhibition and maintain a high firing rate far above the target. Other Pyr cells, which have access to interneurons with a similar stimulus tuning, can recruit inhibition to gradually reduce their firing rates towards the target rate. Because the BCM rule is Hebbian, it tends to strengthen input synapses from Pyr neurons with high activity. This shifts the stimulus tuning of the interneurons to those stimuli that were previously underrepresented. However, this in turn renders a different set of stimuli uncovered by inhibition and withdraws feedback inhibition from the corresponding set of Pyr cells, which can now fire at high rates.

We suspect that this instability can also arise for other Hebbian forms of plasticity in interneuron input synapses when they are combined with homeostatic inhibitory plasticity (*Vogels et al., 2011*) in their output synapses. The underlying reason is that for convergence, the two forms of plasticity need to work synergistically towards the same goal, that is, the same steady state. For two arbitrary synaptic plasticity rules acting in different sets of synapses, it is likely that they aim for two different overall network configurations. Such competition can easily result in latching dynamics with a continuing turn-over of transiently stable states, in which the form of plasticity that acts more quickly gets to reach its goal transiently, only to be undermined by the other one later.

Both Pyr → PV and PV→ Pyr plasticity have been studied in slice (for reviews, see, e.g., *Kullmann and Lamsa, 2007*; *Vogels et al., 2013*), but mostly in isolation. The idea that the two forms of plasticity should act in synergy suggests that it may be interesting to study both forms in the same system, for example, in reciprocally connected Pyr-PV pairs.

Like all computational models, the present one contains simplifying design choices. First, we did not include stimulus-specific *feedforward* inhibition, because the focus lay on the formation of stimulus-specific *feedback* inhibition. The model could be enriched by feedforward inhibition in different ways. In particular, we expect that the two forms of plasticity will establish E/I assemblies even in the presence of stimulus-selective external inputs to the interneurons, because stimulus-specific external excitation should always be more supportive of the homeostatic objective than unspecific inputs. It

may be worth exploring whether adding feedforward inhibition leaves more room for replacing the PV input plasticity that we used by classical Hebbian rules, because the activity of the external inputs remains unaltered by the plasticity in the network (such that the complex instability described above may be mitigated). Given that the focus of this work was on feedback inhibition, an extensive evaluation of the different variants of feedforward inhibition is beyond the scope of the present article.

Second, we neglected much of the complexity of cortical interneuron circuits by including only one class of interneurons. We interpret these interneurons as PV interneurons, given that PV interneurons provide local feedback inhibition (*Hu et al., 2014*) and show a stimulus-selective circuitry akin to E/I assemblies (*Znamenskiy et al., 2018*). With their peri-somatic targets on Pyr cells, PV-expressing (basket) cells are also a prime candidate for the classical feedback model of E/I balance (*van Vreeswijk and Sompolinsky, 1996*). Note that our results do not hinge on any assumptions that are specific to PV neurons and may thus also hold for other interneuron classes that provide feedback inhibition (*Tremblay et al., 2016*). Given that the division of labour of the various cortical interneuron classes is far from understood, an extension to complex interneuron circuits (*Litwin-Kumar et al., 2016*; *Hertäg and Sprekeler, 2019*) is clearly beyond the present study.

Similarly tuned Pyr cells tend to be recurrently connected (*Cossell et al., 2015*; *Harris and Mrsic-Flogel, 2013*), in line with the notion that excitatory cells with similar tuning mutually excite each other. This notion is questioned by a recent perturbation experiment demonstrating feature-specific suppression between Pyr cells with similar tuning (*Chettih and Harvey, 2019*). It has been suggested that this apparently paradoxical effect requires strong and tuned connections between excitatory and inhibitory neurons (*Sadeh and Clopath, 2020*). The E/I assemblies that develop in our model provide sufficiently strong and specific inhibitory feedback to cause a suppression between similarly tuned Pyr neurons in response to perturbations. Hence, despite the presence of stimulus-specific excitatory recurrence, Pyr neurons with similar stimulus preference effectively compete. Computational arguments suggest that this feature competition may be beneficial for stimulus processing, for example, by generating a sparser and more efficient representation of the stimuli (*Olshausen and Field, 2004*; *Denève and Machens, 2016*).

In addition to predicting that knocking out plasticity of inhibitory input or output synapses should prevent the development of E/I assemblies, our model also predicts different outcomes for single-neuron perturbation experiments in juvenile and adult mice. Given that in rodents, stimulus tuning of inhibitory currents occurs later in development than that of excitation (*Dorrn et al., 2010*), we expect that in juvenile mice single-cell perturbations would not cause feature-specific suppression but amplification due to excitatory recurrence and unspecific feedback inhibition.

## Materials and methods

### Network and stimuli

We use custom software to simulate a rate-based recurrent network model containing $N^{\mathrm{E}} = 512$ excitatory and $N^{\mathrm{I}} = 64$ inhibitory neurons. The activation of the neurons follows Wilson-Cowan dynamics:

$$\tau_{\mathrm{E}} \frac{d}{dt} \boldsymbol{h}^{\mathrm{E}} = -\boldsymbol{h}^{\mathrm{E}} + W^{\mathrm{E \leftarrow E}} \boldsymbol{r}^{E} - W^{\mathrm{E \leftarrow I}} \boldsymbol{r}^{\mathrm{I}} + I^{\mathrm{bg}} + \mathbf{I}(\mathbf{s}) \tag{1a}$$

$$\tau_{\mathrm{I}} \frac{d}{dt} \boldsymbol{h}^{\mathrm{I}} = -\boldsymbol{h}^{\mathrm{I}} + W^{\mathrm{I \leftarrow E}} \boldsymbol{r}^{E} - W^{\mathrm{I \leftarrow I}} \boldsymbol{r}^{\mathrm{I}} + I^{\mathrm{bg}} . \tag{1b}$$

Here, $\boldsymbol{r}^{\mathrm{E}} = [\boldsymbol{h}^{\mathrm{E}}]_{+}$, $\boldsymbol{r}^{\mathrm{I}} = [\boldsymbol{h}^{\mathrm{I}}]_{+}$ denote the firing rates of the excitatory and inhibitory neurons, which are given by their rectified activation. $W^{Y \leftarrow X}$ denotes the matrix of synaptic efficacies from population $X$ to population $Y$ ($X, Y \in \{\mathrm{E, I}\}$). The external inputs $\mathbf{I}(\mathbf{s})$ to the excitatory neurons have a bell-shaped tuning in the three-dimensional stimulus space consisting of spatial frequency, temporal frequency, and orientation (*Znamenskiy et al., 2018*). To avoid edge effects, the stimulus space is periodic in all three dimensions, with stimuli ranging from $-\pi$ to $\pi$. The stimulus tuning of the external inputs is modelled by a von Mises function with a maximum of 50 Hz and a tuning width $\kappa = 1$. The preferred stimuli of the $N^{\mathrm{E}} = 512$ excitatory cells cover the stimulus space evenly on a $12 \times 12 \times 12$ grid. All neurons receive a constant background input of $I^{\mathrm{bg}} = 5$ Hz.

Recurrent connections $W^{\text{E}\leftarrow\text{E}}$ among excitatory neurons have synaptic weight between neurons $i$ and $j$ that grows linearly with the signal correlation of their external inputs:

$$W^{\text{E}\leftarrow\text{E}}_{ij} = \left[ corr(I_i(\mathbf{s}), I_j(\mathbf{s})) - C \right]_+ . \tag{2}$$

The cropping threshold $C$ is chosen such that the overall connection among the excitatory neurons probability is 0.6. The remaining synaptic connections (E→I, I→E, I→I) are initially random, with a connection probability $p = 0.6$ and log-normal weights. For parameters, please refer to *Table 1*.

During learning, we repeatedly draw all $12 \times 12 \times 12$ preferred stimuli of the Pyr neurons, in random order. This procedure is repeated 500 times to ensure convergence of synaptic weights. To reduce simulation time, we present each stimulus long enough for all firing rates to reach steady state and only then update the synaptic weights.

## Synaptic plasticity

The PV → Pyr and Pyr → PV synapses follow plasticity rules that aim to minimise the deviation of the excitatory activations from a target rate $\rho_0$ ($\rho_0 = 1$ Hz):

$$\mathcal{E}(\boldsymbol{h}^{\text{E}}) = \left\langle \frac{1}{2} \sum_{j=1}^{N^{\text{E}}} \left( h^{\text{E}}_j - \rho_0 \right)^2 \right\rangle_{\mathbf{s}} , \tag{3}$$

where $\langle \cdot \rangle_{\mathbf{s}}$ denotes the average over all stimuli. When plastic, synaptic weights change according to

$$\Delta W^{\text{E}\leftarrow\text{I}}_{ji} \propto \left( h^{\text{E}}_j - \rho_0 \right) r^{\text{I}}_i , \tag{4a}$$

$$\Delta W^{\text{I}\leftarrow\text{E}}_{ij} \propto \left[ \sum_{k=1}^{N^{\text{E}}} W^{\text{I}\leftarrow\text{E}}_{ik} \left( h^{\text{E}}_k - \rho_0 \right) \right] r^{\text{E}}_j \tag{4b}$$

$$\approx \left[ \sum_{k=1}^{N^{\text{E}}} W^{\text{I}\leftarrow\text{E}}_{ik} \left( r^{\text{E}}_k - \rho_0 \right) \right] r^{\text{E}}_j$$

$$= \left( I^{E,rec}_i - I_0 \right) r^{\text{E}}_j . \tag{4c}$$

After every update of the Pyr → PV matrix, the incoming weights for each PV interneuron are

**Table 1.** Model parameters.

| $N^{\text{E}}$ | **512** | $N^{\text{I}}$ | **64** | **Number of exc. and inh. neurons.** |
|---|---|---|---|---|
| $\tau_{\text{E}}$ | 50 ms | $\tau_{\text{I}}$ | 25 ms | Rate dynamics time constants |
| $dt$ | 1 ms | | | Numerical integration time step |
| $p^{\text{E}\leftarrow X}$ | 0.6 | $p^{\text{I}\leftarrow X}$ | 0.6 | Connection probability to exc. and inh. neurons |
| $J^{\text{E}\leftarrow\text{E}}_i$ | 2 | $J^{\text{I}\leftarrow\text{E}}_i$ | 5 | Total of exc. weights onto neuron $i$: $\sum_j W^{X\leftarrow\text{E}}_{ij}$ |
| $J^{\text{E}\leftarrow\text{I}}_i$ | 1 | $J^{\text{I}\leftarrow\text{I}}_i$ | 1 | Total of inh. weights onto neuron $i$: $\sum_j W^{X\leftarrow\text{I}}_{ij}$ |
| $\sigma^{\text{E}\leftarrow X}$ | 0.65 | $\sigma^{\text{I}\leftarrow X}$ | 0.65 | Std. deviation of the logarithm of the weights |
| $\theta^{\text{E}\leftarrow\text{I}}$ | $10^{-4}$ | $\theta^{\text{I}\leftarrow\text{E}}$ | $10^{-4}$ | Experimental detection threshold for synapses |
| $I^{\text{bg}}$ | 5 Hz | $\max(\mathbf{I}(\mathbf{s}))$ | 50 Hz | Background and maximum stimulus-specific input |
| $N^S$ | $12 \times 12 \times 12$ | $N^{\text{trials}}$ | 500 | Number of stimuli and trials |
| $R^S$ | $2\pi \times 2\pi \times 2\pi$ | $\kappa$ | 1 | Range of stimuli and Pyr RF von Mises width |
| $\Delta I$ | 10 Hz | | | Change of input for perturbation experiments |
| $\eta^{\text{Approx.}}$ | $10^{-5}$ | $\eta^{\text{Grad.}}$ | $10^{-3}$ | Learning rates (approximate and gradient rules) |
| $\delta^{\text{E}\leftarrow\text{I}}$ | 0.1 | $\delta^{\text{I}\leftarrow\text{E}}$ | 0.1 | Weight decay rates |
| $\rho_0$ | 1 Hz | | | Homeostatic plasticity target |
| $\beta_1$ | 0.9 | $\beta_2$ | 0.999 | Adam parameters for gradient rules |
| $\epsilon$ | $10^{-9}$ | | | |

multiplicatively scaled such that their sum is $J^{\mathrm{I \leftarrow E}}$ (**Akrout et al., 2019**). In that case, the rule in **Equation 4**b is approximately local in that it compares the excitatory input current $I_i^{E,\mathrm{rec}}$ received by the postsynaptic PV neuron to a target value $I_0 = J^{\mathrm{I \leftarrow E}} \rho_0$, and adjusts the incoming synapses in proportion to this error and to presynaptic activity (see **Equation 4**c).

Both plasticity rules are approximations of the gradient of the objective function **Equation 3**. Interested readers are referred to Appendix 1 for their mathematical derivation. For the results in **Figure 1—figure supplement 2**, we use the Adaptive Moment Estimation (Adam) algorithm (**Kingma and Ba, 2014**) to improve optimisation performance.

We used a standard reparameterisation method to ensure the sign constraints of an E/I network. Moreover, all weights are subject to a small weight-dependent decay term, which aids to keep the firing rates of the interneurons in a reasonable range. For details, please refer to Appendix 1 . The learning rule **Equation 4**a for the output synapses of the inhibitory neurons is similar to the rule proposed by **Vogels et al., 2011**, wherein each inhibitory synapse increases in strength if the deviation of the postsynaptic excitatory cell from the homeostatic target $\rho_0$ is positive (and decreases it when negative). In contrast, the learning rule **Equation 4**b increases activated input synapses for an interneuron if the weighted sum of deviations in its presynaptic excitatory population is positive (and decreases them if it is negative). Though it is local, when operating in conjunction with the plasticity of **Equation 4**a, this leads to feedback alignment in our simulations and effectively performs backpropagation without the need for weight transport (**Akrout et al., 2019**).

Note that the objective function **Equation 3** can also be interpreted differently. The activation $h^{\mathrm{E}}$ of a neuron is essentially the difference between its excitatory and inhibitory inputs. Therefore, the objective function **Equation 3** is effectively the mean squared error between excitation and inhibition, aside from a small constant offset $\rho_0$. The derived learning rules can therefore be seen as supervised learning of the inhibitory inputs, with excitation as the label. They hence aim to establish the best co-tuning of excitation and inhibition that is possible given the circuitry.

## Perturbation experiments

The perturbation experiments in **Figure 4** are performed in a network in which both forms of plasticity have converged. The network is then exposed to different stimuli, while the afferent drive to a single excitatory cell $i$ is transiently increased by $\Delta I = 10$ Hz. For each stimulus, we compute the steady-state firing rates $r_j$ of all excitatory cells both with and without the perturbation. The influence of the perturbation of neuron $i$ on neuron $j$ is defined as the difference between these two firing rates, normalised by the pertubation magnitude (**Sadeh and Clopath, 2020**). This stimulation protocol is repeated for 90 randomly selected excitatory neurons. The dependence of the influence on the tuning similarity (**Figure 4d**) is obtained by binning the influence of the perturbed neuron $i$ and the influenced neuron $j$ according to their stimulus response correlation, and then averaging across all influences in the bin. During the perturbation experiments, synaptic plasticity was disabled.

## Quantitative measures

The response similarity (RS) of the stimulus tuning of two neurons $i$ and $j$ is measured by the dot product of their steady-state firing rates in response to all stimuli, normalised by the product of their norms (**Znamenskiy et al., 2018**):

$$\mathrm{RS}(r_i, r_j) = \frac{\sum_{\mathbf{s}} r_i(\mathbf{s}) r_j(\mathbf{s})}{\left(\sum_{\mathbf{s}} (r_i(\mathbf{s}))^2 \sum_{\mathbf{s}} (r_j(\mathbf{s}))^2\right)^{1/2}}. \tag{5}$$

The same measure is used for the similarity of synaptic currents onto excitatory neurons in **Figure 1—figure supplement 3c** and **Figure 1—figure supplement 2d**.

There is no structural plasticity, that is, synapses are never added or pruned. However, when calculating Pearson's correlation between synaptic weights and RS, we exclude synapses that are too weak to be detected using the experimental protocol employed by **Znamenskiy et al., 2018**. The threshold values $\theta^{\mathrm{E \leftarrow I}}$ and $\theta^{\mathrm{I \leftarrow E}}$ were chosen to be approximately four orders of magnitude weaker than the strongest synapses in the network. The rules that we investigate here tend to produce bimodal distributions of weights, with the lower mode well below this threshold (**Figure 1—figure supplement 5**).

The stimulus selectivity of the neurons is measured by the skewness of their response distribution across all stimuli:

$$\gamma_i = \frac{\left\langle (r_i(\mathbf{s}) - \bar{r}_i)^3 \right\rangle_{\mathbf{s}}}{\left\langle (r_i(\mathbf{s}) - \bar{r}_i)^2 \right\rangle_{\mathbf{s}}^{3/2}} \tag{6}$$

where $\bar{r}_i = \langle r_i(\mathbf{s}) \rangle_{\mathbf{s}}$. Both the RS *Equation 5* and the stimulus selectivity *Equation 6* are adapted from *Znamenskiy et al., 2018*.

Finally, the angle $\theta$ between the gradient $G$ from *Equation 15* and its approximation $A$ from *Equation 4* is given by

$$\theta = \arccos\left( \frac{\sum_{ij} G_{ij} A_{ij}}{\left( \sum_{ij} G_{ij}^2 \sum_{ij} A_{ij}^2 \right)^{1/2}} \right) \tag{7}$$

## Acknowledgements

We thank Joram Keijser for helpful discussions that inspired parts of this work. He, along with Denis Alevi, Loreen Hertäg and Robert T Lange also provided careful proof-reading of the manuscript. This project was funded by the German Federal Ministry for Science and Education through a Bernstein Award (BMBF, FKZ 01GQ1201) and by the German Research Foundation (DFG, collaborative research centre FOR 2143).

## Additional information

### Funding

| Funder | Grant reference number | Author |
| --- | --- | --- |
| Bundesministerium für Bildung und Forschung | FKZ 01GQ1201 | Owen Mackwood Henning Sprekeler |
| Deutsche Forschungsgemeinschaft | FOR 2143 | Owen Mackwood Henning Sprekeler |

The funders had no role in study design, data collection and interpretation, or the decision to submit the work for publication.

### Author contributions

Owen Mackwood, Conceptualization, Data curation, Software, Formal analysis, Investigation, Visualization, Methodology, Writing - original draft, Writing - review and editing; Laura B Naumann, Writing - original draft, Project administration, Writing - review and editing; Henning Sprekeler, Conceptualization, Resources, Formal analysis, Supervision, Funding acquisition, Writing - original draft, Project administration, Writing - review and editing

### Author ORCIDs

Owen Mackwood https://orcid.org/0000-0001-8569-6614
Laura B Naumann https://orcid.org/0000-0002-7919-7349
Henning Sprekeler https://orcid.org/0000-0003-0690-3553

### Decision letter and Author response

Decision letter https://doi.org/10.7554/eLife.59715.sa1
Author response https://doi.org/10.7554/eLife.59715.sa2

## Additional files

### Supplementary files

- Transparent reporting form

## Data availability

Source code for the simulator that generated all data is available at https://github.com/owenmack-wood/ei-assemblies (copy archived at https://archive.softwareheritage.org/swh:1:rev:e2f029a7e7285230cbbdbc7e817e25c8c5535fc1).

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

# Appendix 1

## Plasticity rules

The general framework we follow to derive homeostatic rules is to minimise the mean squared deviation of individual excitatory (Pyr) neuron activations from a target for all stimuli. More specifically, we perform gradient descent on the following objective function:

$$\mathcal{E}(\boldsymbol{h}^{\mathrm{E}}) = \left\langle \frac{1}{2} \sum_{j=1}^{N^{\mathrm{E}}} \left( h_j^{\mathrm{E}} - \rho_0 \right)^2 \right\rangle_{\mathbf{s}}.$$

Note that the activations $\boldsymbol{h}^{\mathrm{E}}$ are given by the difference between the excitatory and the inhibitory inputs to the excitatory neurons. Our approach can hence be interpreted as supervised learning of the inhibitory circuitry, with the goal of minimising the mean squared loss between the inhibitory and the excitatory inputs (plus the constant target $\rho_0$). In this sense, the derived gradient rules aim to generate the best possible E/I balance across stimuli that is possible with the circuitry at hand.

For reasons of readability, we will first simply state the derived rules. The details of their derivation can be found in the following section.

The sign constraints in excitatory-inhibitory networks require all synaptic weights to remain positive. To ensure this, we reparameterised all plastic weights of the network by a strictly positive softplus function $W = s^+(V) = \alpha^{-1} \ln(1 + \exp \alpha V)$ and optimised the weight parameter $V$ by gradient descent.

In summary, the derived learning rules for the synaptic weight parameters between excitatory neuron $j$ and inhibitory interneuron $i$ are given by

$$\Delta V_{ji}^{\mathrm{E \leftarrow I}} = \eta^{\mathrm{I}} \left( h_j^{\mathrm{E}} - \rho_0 \right) \frac{\partial W_{ji}^{\mathrm{E \leftarrow I}}}{\partial V_{ji}^{\mathrm{E \leftarrow I}}} r_i^{\mathrm{I}} - \delta^{\mathrm{I}} W_{ji}^{\mathrm{E \leftarrow I}}, \tag{8a}$$

$$\Delta V_{ij}^{\mathrm{I \leftarrow E}} = \eta^{\mathrm{E}} \left[ \sum_{k=1}^{N^{\mathrm{E}}} W_{ik}^{\mathrm{I \leftarrow E}} \left( h_k^{\mathrm{E}} - \rho_0 \right) \right] \frac{\partial r_i^{\mathrm{I}}}{\partial h_i^{\mathrm{I}}} \frac{\partial W_{ij}^{\mathrm{I \leftarrow E}}}{\partial V_{ij}^{\mathrm{I \leftarrow E}}} r_j^{\mathrm{E}} - \delta^{\mathrm{E}} W_{ij}^{\mathrm{I \leftarrow E}}. \tag{8b}$$

Please note that we added a small weight decay to both learning rules. The purpose of this decay term is to avoid an ambiguity in the solution. When the firing rates of the interneurons are increased, but their output weights are decreased accordingly, the firing rates of the excitatory population remain unchanged. Pure gradient-based rules can therefore generate extreme values for the synaptic weights, in which the interneurons have biologically unrealistic firing rates. The additional decay terms in the learning rules solve this issue.

Finally, we replaced the derivative $\frac{\partial r}{\partial h}$ (which should be a Heaviside function, because rates are the rectified activations) by the derivative of a soft-plus function with finite sharpness ($\alpha = 1$). This allows interneurons to recover from a silent state, in which all gradients vanish. Note that this replacement is done only in the learning rules. The firing rates are still the rectified activations. This method is similar to recent surrogate gradient approaches in spiking networks (***Neftci et al., 2019***).

## Derivation of the homeostatic plasticity rules in recurrent networks

The challenging aspect of the derivation of the learning rules lies in the recurrence of the network. The effects of changes in individual synapses can percolate through the network and thereby change the firing rates of all neurons. Moreover, the temporal dynamics of the network would in principle require a backpropagation of the gradient through time. We circumvent this complication by assuming that the external stimuli to the network change slowly compared to the dynamical time scales of the network, and that the network adiabatically follows the fixed point in its dynamics as the stimulus changes. This assumption significantly simplifies the derivation of the gradient.

The goal is to minimise the total deviation of the excitatory activations $\boldsymbol{h}^{\mathrm{E}}$ from the homeostatic target value $\rho_0$. To this end, we calculate the gradient of the objective function in ***Equation 3*** with respect to a given synaptic weight parameter $v \in \{V_{ij}^{\mathrm{I \leftarrow E}}, V_{ji}^{\mathrm{E \leftarrow I}}\}$:

$$\frac{\partial}{\partial v}\mathcal{E}(\boldsymbol{h}^{\mathrm{E}}) = \left\langle \left(\boldsymbol{h}^{\mathrm{E}} - \rho_0\right)^{\top}\frac{\partial \boldsymbol{h}^{\mathrm{E}}}{\partial v}\right\rangle_{\mathbf{s}}. \tag{9}$$

We therefore need the gradient of the activations $\boldsymbol{h}^{\mathrm{E}}$ of excitatory cells with respect to a parameter $v$. In the steady state, the activations are given by

$$\boldsymbol{h}^{\mathrm{E}} = W^{\mathrm{E}\leftarrow\mathrm{E}}\boldsymbol{r}^{\mathrm{E}} - W^{\mathrm{E}\leftarrow\mathrm{I}}\boldsymbol{r}^{\mathrm{I}} + I_{\mathrm{bg}} + \mathbf{I}(\mathbf{s}). \tag{10}$$

The gradient of the activations $\boldsymbol{h}^{\mathrm{E}}$ is therefore given by the following implicit condition:

$$\frac{\partial \boldsymbol{h}^{\mathrm{E}}}{\partial v} = W^{\mathrm{E}\leftarrow\mathrm{E}}D^{\mathrm{E}}\frac{\partial \boldsymbol{h}^{\mathrm{E}}}{\partial v} - \left[\frac{\partial W^{\mathrm{E}\leftarrow\mathrm{I}}}{\partial v}\boldsymbol{r}^{\mathrm{I}} + W^{\mathrm{E}\leftarrow\mathrm{I}}D^{\mathrm{I}}\frac{\partial \boldsymbol{h}^{\mathrm{I}}}{\partial v}\right], \tag{11}$$

where we introduced the diagonal matrices $D_{ij}^{\mathrm{E/I}} := \delta_{ij}\partial r_i^{\mathrm{E/I}}/\partial h_i^{\mathrm{E/I}}$ for notational convenience, $\delta_{ij}$ being the Kronecker symbol. Derivatives of expressions that do not depend on any of the synaptic weights in question are excluded.

*Equation 11* requires the gradient $\frac{\partial \boldsymbol{h}^{\mathrm{I}}}{\partial v}$ of the inhibitory activations with respect to the parameter $v$, which can be calculated by a similar approach:

$$\begin{aligned}\frac{\partial \boldsymbol{h}^{\mathrm{I}}}{\partial v} &= \frac{\partial}{\partial v}\left(W^{\mathrm{I}\leftarrow\mathrm{E}}\boldsymbol{r}^{\mathrm{E}} - W^{\mathrm{I}\leftarrow\mathrm{I}}\boldsymbol{r}^{\mathrm{I}} + I_{\mathrm{bg}}\right) \\ &= \left(\frac{\partial W^{\mathrm{I}\leftarrow\mathrm{E}}}{\partial v}\boldsymbol{r}^{\mathrm{E}} + W^{\mathrm{I}\leftarrow\mathrm{E}}D^{\mathrm{E}}\frac{\partial \boldsymbol{h}^{\mathrm{E}}}{\partial v}\right) - W^{\mathrm{I}\leftarrow\mathrm{I}}D^{\mathrm{I}}\frac{\partial \boldsymbol{h}^{\mathrm{I}}}{\partial v}.\end{aligned}$$

Introducing the effective interaction matrix $\mathcal{M} := \mathbb{I} + W^{\mathrm{I}\leftarrow\mathrm{I}}D^{\mathrm{I}}$ among the interneurons ($\mathbb{I}$ being the identity matrix) allows to solve for the gradient of $\boldsymbol{h}^{\mathrm{I}}$:

$$\frac{\partial \boldsymbol{h}^{\mathrm{I}}}{\partial v} = \mathcal{M}^{-1}\left[W^{\mathrm{I}\leftarrow\mathrm{E}}D^{\mathrm{E}}\frac{\partial \boldsymbol{h}^{\mathrm{E}}}{\partial v} + \frac{\partial W^{\mathrm{I}\leftarrow\mathrm{E}}}{\partial v}\boldsymbol{r}^{\mathrm{E}}\right]$$

Inserting this expression into *Equation 11* yields

$$\frac{\partial \boldsymbol{h}^{\mathrm{E}}}{\partial v} = \left[W^{\mathrm{E}\leftarrow\mathrm{E}}D^{\mathrm{E}} - W^{\mathrm{E}\leftarrow\mathrm{I}}D^{\mathrm{I}}\mathcal{M}^{-1}W^{\mathrm{I}\leftarrow\mathrm{E}}D^{\mathrm{E}}\right]\frac{\partial \boldsymbol{h}^{\mathrm{E}}}{\partial v} - \frac{\partial W^{\mathrm{E}\leftarrow\mathrm{I}}}{\partial v}\boldsymbol{r}^{\mathrm{I}} - W^{\mathrm{E}\leftarrow\mathrm{I}}D^{\mathrm{I}}\mathcal{M}^{-1}\frac{\partial W^{\mathrm{I}\leftarrow\mathrm{E}}}{\partial v}\boldsymbol{r}^{\mathrm{E}}.$$

Introducing the effective interaction matrix $\mathcal{W} = \mathbb{I} - W^{\mathrm{E}\leftarrow\mathrm{E}}D^{\mathrm{E}} + W^{\mathrm{E}\leftarrow\mathrm{I}}D^{\mathrm{I}}\mathcal{M}^{-1}W^{\mathrm{I}\leftarrow\mathrm{E}}D^{\mathrm{E}}$ among the excitatory neurons yields an explicit expression for the gradient of $\boldsymbol{h}^{\mathrm{E}}$:

$$\frac{\partial \boldsymbol{h}^{\mathrm{E}}}{\partial v} = -\mathcal{W}^{-1}\frac{\partial W^{\mathrm{E}\leftarrow\mathrm{I}}}{\partial v}\boldsymbol{r}^{\mathrm{I}} - \mathcal{W}^{-1}W^{\mathrm{E}\leftarrow\mathrm{I}}D^{\mathrm{I}}\mathcal{M}^{-1}\frac{\partial W^{\mathrm{I}\leftarrow\mathrm{E}}}{\partial v}\boldsymbol{r}^{\mathrm{E}}. \tag{12}$$

To obtain gradients with respect to a particular network parameter, we simply substitute the chosen parameter into *Equation 12*. For the parameters $V_{ij}^{\mathrm{I}\leftarrow\mathrm{E}}$ of the input synapses to the interneurons, the gradient reduces to

$$\frac{\partial \boldsymbol{h}^{\mathrm{E}}}{\partial V^{\mathrm{I}\leftarrow\mathrm{E}}} = -\mathcal{W}^{-1}W^{\mathrm{E}\leftarrow\mathrm{I}}D^{\mathrm{I}}\mathcal{M}^{-1}\frac{\partial W^{\mathrm{I}\leftarrow\mathrm{E}}}{\partial V^{\mathrm{I}\leftarrow\mathrm{E}}}\boldsymbol{r}^{\mathrm{E}}, \tag{13}$$

and for the parameters $V_{ij}^{\mathrm{E}\leftarrow\mathrm{I}}$ of the output synapses from the interneurons we get

$$\frac{\partial \boldsymbol{h}^{\mathrm{E}}}{\partial V^{\mathrm{E}\leftarrow\mathrm{I}}} = -\mathcal{W}^{-1}\frac{\partial W^{\mathrm{E}\leftarrow\mathrm{I}}}{\partial V^{\mathrm{E}\leftarrow\mathrm{I}}}\boldsymbol{r}^{\mathrm{I}}. \tag{14}$$

By inserting these expressions into *Equation 9* and dropping the average, we obtain online learning rules for the input and output synapses of the interneurons:

$$\Delta V^{\mathrm{I}\leftarrow\mathrm{E}} \propto \left[\left(\boldsymbol{h}^{\mathrm{E}} - \rho_0\right)^{\top}\mathcal{W}^{-1}W^{\mathrm{E}\leftarrow\mathrm{I}}D^{\mathrm{I}}\mathcal{M}^{-1}\right]\frac{\partial W^{\mathrm{I}\leftarrow\mathrm{E}}}{\partial V^{\mathrm{I}\leftarrow\mathrm{E}}}\boldsymbol{r}^{\mathrm{E}} \tag{15a}$$

$$\Delta V^{\mathrm{E}\leftarrow\mathrm{I}} \propto \left[\left(\boldsymbol{h}^{\mathrm{E}} - \rho_0\right)^{\top}\mathcal{W}^{-1}\right]\frac{\partial W^{\mathrm{E}\leftarrow\mathrm{I}}}{\partial V^{\mathrm{E}\leftarrow\mathrm{I}}}\boldsymbol{r}^{\mathrm{I}}. \tag{15b}$$

Note that the same approach also yields learning rules for the threshold and the gain of the transfer function of the inhibitory interneurons, if those are parameters of the system. Although we did not use such intrinsic plasticity rules, we include them here for the interested reader. We assumed a threshold linear transfer function of the interneurons: $r_i^{\mathrm{I}} = g_i [h_i^{\mathrm{I}} - \theta_i]^+$, where $g_i$ is the gain of the neuronal transfer function and $\theta_i$ a firing threshold. While the firing threshold can become negative, gain is reparameterised via the strictly positive soft-plus $g_i = s^+(v_i^g)$. The gradient-based learning rule for the firing thresholds $\theta_i$ of the interneurons is given by

$$\Delta\theta_i \propto -\left[\left(\boldsymbol{h}^{\mathrm{E}} - \rho_0\right)^\top \mathcal{W}^{-1} W^{\mathrm{E}\leftarrow\mathrm{I}} \mathcal{M}^{-1}\right]_i \frac{\partial r_i^{\mathrm{I}}}{\partial\theta_i}, \tag{16}$$

and the corresponding learning rule for the interneuron gain $g_i$ is

$$\Delta v_i^g \propto -\left[\left(\boldsymbol{h}^{\mathrm{E}} - \rho_0\right)^\top \mathcal{W}^{-1} W^{\mathrm{E}\leftarrow\mathrm{I}} \mathcal{M}^{-1}\right]_i \frac{\partial r_i^{\mathrm{I}}}{\partial g_i} \frac{\partial g_i}{\partial v_i^g}. \tag{17}$$

## Approximating the gradient rules

In the gradient-based rules derived in the previous section, the $\mathcal{W}^{-1}$ and $\mathcal{M}^{-1}$ terms account for the fact that a change in a given synaptic connections percolates through the network. As a result, the learning rules are highly non-local and hard to implement in a biologically plausible way. To resolve this challenge, we begin by noting that

$$\mathcal{W}^{-1} = \left(\mathbb{I} - \hat{\mathcal{W}}\right)^{-1} = \sum_{k=0}^{\infty} \hat{\mathcal{W}}^k,$$

which holds if $\left\|\hat{\mathcal{W}}\right\| < 1$. $\hat{\mathcal{W}}$ is a matrix that depends on the synaptic weights in the network. A similar relation holds for $\mathcal{M}^{-1}$. Since those matrices are contained in *Equation 15*a, we substitute the equivalent sums into the relevant sub-expression and truncate the geometric series after the zeroth order, as in

$$
\begin{aligned}
\mathcal{W}^{-1} W^{\mathrm{E}\leftarrow\mathrm{I}} D^{\mathrm{I}} \mathcal{M}^{-1} &= \left(\sum_{k=0}^{\infty} \hat{\mathcal{W}}^k\right) W^{\mathrm{E}\leftarrow\mathrm{I}} D^{\mathrm{I}} \left(\sum_{k=0}^{\infty} \hat{\mathcal{M}}^k\right) \\
&= W^{\mathrm{E}\leftarrow\mathrm{I}} D^{\mathrm{I}} + \hat{\mathcal{W}} W^{\mathrm{E}\leftarrow\mathrm{I}} D^{\mathrm{I}} + W^{\mathrm{E}\leftarrow\mathrm{I}} D^{\mathrm{I}} \hat{\mathcal{M}} + \left(\sum_{k=1}^{\infty} \hat{\mathcal{W}}^k\right) W^{\mathrm{E}\leftarrow\mathrm{I}} D^{\mathrm{I}} \left(\sum_{k=1}^{\infty} \hat{\mathcal{M}}^k\right) \\
&\approx W^{\mathrm{E}\leftarrow\mathrm{I}} D^{\mathrm{I}}.
\end{aligned}
$$

The truncation to zeroth order in the last line should yield an acceptable approximation if synapses are sufficiently weak. The effect of higher-order interactions in the network can then be ignored. This approximation can be substituted into *Equation 15*a and yields an equation that resembles a backpropagation rule in a feedforward network (E → I → E) with one hidden layer—the interneurons. The final, local approximation used for the simulations in the main text is then reached by replacing the output synapses of the interneurons by the transpose of their input synapses. While there is no mathematical argument why this replacement is valid, it turns out to be in the simulations, presumably because of a mechanism akin to feedback alignment (*Lillicrap et al., 2016*; see discussion in the main text). In feedback alignment, the matrix that backpropagates the errors is replaced by a random matrix $B$. Here, we instead use the feedforward weights in the layer below. Similar to the extension to feedback alignment of *Akrout et al., 2019*, those weights are themselves plastic. However, we believe that the underlying mechanism of feedback alignment still holds. The representation in the hidden layer (the interneurons) changes as if the weights to the output layer (the Pyr neurons) were equal to the weight matrix they are replaced with (here, the input weights to the PV neurons). To exploit this representation, the weights to the output layer then align to the replacement weights, justifying the replacement post hoc (*Figure 1G*).

Note that the condition for feedback alignment to provide an update in the appropriate direction ($e^T B^T W e > 0$, where $e$ denotes the error, $W$ the weights in the second layer, and $B$ the random feedback matrix) reduces to the condition that $W^{\mathrm{E}\leftarrow\mathrm{I}} W^{\mathrm{I}\leftarrow\mathrm{E}}$ is positive definite (assuming the errors are full rank). One way of assuring this is a sufficiently positive diagonal of this matrix product, that is, a

sufficiently high correlation between the incoming and outgoing synapses of the interneurons. A positive correlation of these weights is one of the observations of *Znamenskiy et al., 2018* and also a result of learning in our model.

While such a positive correlation is not necessarily present for all learning tasks or network models, we speculate that it will be for the task of learning an E/I balance in networks that obey Dale's law.

The same logic of using a zeroth order approximation of $\mathcal{W}^{-1}$ that neglects higher-order interactions is employed to recover the inhibitory synaptic plasticity rule of *Vogels et al., 2011* from *Equation 15b*.

Overall, the local approximation of the learning rule relies on three assumptions: slowly varying inputs, weak synaptic weights, and alignment of input and output synapses of the interneurons. These assumptions clearly limit the applicability of the learning rules for other learning tasks. In particular, the learning rules will not allow the network to learn temporal sequences.

## The effect of static but heavy-tailed PV input weights

We investigated whether plasticity is required on both input and output synapses. The main argument why plasticity may not be required is that a static but heavy-tailed weight distribution for Pyr → PV synapses might provide sufficient stimulus selectivity in PV neurons, such that plasticity of output synapses alone can account for the experimental data of *Znamenskiy et al., 2018*. To test this, we sampled weights from a log-normal distribution (*Loewenstein et al., 2011*), with parameters that are in line with the data reported by *Znamenskiy et al., 2018*. Weights spanned approximately two orders of magnitude. For this setting, PV neurons do not have sufficient stimulus selectivity for output plasticity alone to co-tune inhibitory currents with excitatory currents, when Pyr → PV plasticity is knocked out. Moreover, if we sample PV input and output synapses, correlations between excitatory input synapse strength and RS are not reliably found, nor are correlations between input and output synapse strength (*Figure 3*).

Nevertheless, one could expect that a sufficiently heavy-tailed distribution of PV excitatory input weights could account for the experimentally observed correlations, even in the absence of Pyr → PV plasticity. We therefore repeated the simulations of *Figure 3* with a weight distribution that spans almost five orders of magnitude (*Figure 3—figure supplement 2*). We also tested other distributions, but the results remain qualitatively unchanged.

With this heavy-tailed distribution of excitatory input weights, PV neurons indeed exhibit increased stimulus selectivity (*Figure 3—figure supplement 2b*, cf. *Figure 3c*). This increased selectivity enables greater stimulus-specific co-tuning of inhibitory and excitatory currents (*Figure 3—figure supplement 2c*, cf. *Figure 1—figure supplement 2d*). Despite this, repeated random sampling of excitatory input synaptic weights ($n = 100$ drawn $10^4$ times) produces positive correlations with RS only about half the time, and for reciprocally connected Pyr-PV cell pairs, input synaptic weights are unlikely to be correlated with output weights (*Figure 3—figure supplement 2d*). Observing the correlations of *Znamenskiy et al., 2018* is therefore unlikely in the absence of some kind of PV input plasticity.

Note that the setting in *Figure 3—figure supplement 2* is conservative in that we chose parameters that made the observation of positive correlations likely. In particular, it is important to note that—similar to the results of *Figure 3*—we consider synapses below a chosen threshold as too weak to be detected during the sampling procedure. We set the threshold of detectability to be approximately two orders of magnitude below the strongest weights (dashed lines in *Figure 3—figure supplement 2e*). For lower thresholds, the fraction of significant Exc-RS and Exc-Inh correlations rapidly decreases, to essentially zero if we include all weights. This is due to the fact that the inhibitory stimulus tuning is determined by a small number of large weights, and if samples are drawn from all synapses, the probability of sampling one of these large weights is small.

In summary, randomly drawn input weights to PV neurons from the distribution observed by *Znamenskiy et al., 2018* are not sufficiently sparse to ensure enough PV selectivity for the observed E/I assemblies, and sampling from even sparser distributions makes the observation of the few influential weights too unlikely. This suggests that the input weights to PV neurons are not random, but that these synaptic weights and the stimulus tuning of their inputs are correlated, consistent with the presence of synaptic plasticity.

Of course, the question whether random connectivity is sufficient for selectivity not only depends on the weight distribution, but also on the total number of synapses received by the PV neurons and the stimulus selectivity of the Pyr neurons they receive input from. Even when the weights are not sparsely distributed, a small number of inputs combined with sufficiently sparse inputs could generate substantial selectivity. We did not explore these additional parameters (in-degree of PV neurons, Pyr selectivity) in detail here, because we believe that our choices are already rather conservative. Due to the small size of the network, PV neurons receive only about 300 excitatory inputs, which is unlikely to be an overestimate, and the stimulus tuning of the Pyr neurons in the model is comparable to V1 (*Znamenskiy et al., 2018*, *Figure 1*), although the diversity of the Pyr neuron selectivity in the data is of course higher.

