## [Decision Letter]

**Acceptance summary:**

Understanding the connectivity patterns observed in the brain and how these connectivity patterns emerge from local, cell-to-cell plasticity is a grand challenge in modern neuroscience. This manuscript describes conditions under which synaptic plasticity can organize excitatory/inhibitory networks in the brain into assemblies in which both kinds of neurons have structured connectivity, as observed in real networks. The work makes predictions about what kinds of plasticity can give rise to this and accounts for recent experimental findings about manipulating inhibitory neurons and, thereby, the stimulus-dependence of their connectivity. It will be of interest to both the experimental and theoretical neuroscience community, and asks and answers a deep question about how the brain self-organizes its detailed connectivity patterns.

**Decision letter after peer review:**

[Editors’ note: the authors submitted for reconsideration following the decision after peer review. What follows is the decision letter after the first round of review.]

Thank you for submitting your work entitled "Learning excitatory-inhibitory neuronal assemblies in recurrent networks" for consideration by *eLife*. Your article has been reviewed by two peer reviewers, one of whom is a member of our Board of Reviewing Editors, and the evaluation has been overseen by a Senior Editor. The reviewers have opted to remain anonymous.

Our decision has been reached after consultation between the reviewers. Based on these discussions and the individual reviews below, we regret to inform you that your work will not be considered further for publication in *eLife*.

The work detailed here explores model of recurrent cortical networks and shows that synaptic plasticity must be present in both excitatory to inhibitory neurons and vice versa to produce the known E/I assemblies found in cortex. There are some interesting findings about the consequences of assemblies formed in this way. A major claim in the manuscript (that argues for the broad impact of the work) is that this shows for the first time how a local approximation rule can instantiate feedback in a biologically plausible way.

While the reviewers found the work to be solid and interesting, they failed to find that the work was appropriate for *eLife*, specifically because of other recent papers that show that a biologically plausible alternative to backpropagation can be instantiated in recurrent neural nets, e.g. a paper published here last year by J. Murray, Local online learning in recurrent networks with random feedback. It's understood that the authors were focusing here on the E/I interactions, but in that case it seems that the novelty of the result needs to be somewhat reframed.

The reviewers were also concerned about the exposition in the introduction and some results that could have been added to a few figures, and had questions about why exactly a BCM rule did not work in this model. Those technical concerns along with doubts about the strong novelty claim led to this decision.

Reviewer #1:

The manuscript investigates the situations in which stimulus-specific assemblies can emerge in a recurrent network of excitatory (E) and inhibitory (I, presumed parvalbumin-positive) neurons. The authors combine (1) Hebbian plasticity of I->E synapses that is proportional to the difference between the E neuron's firing rate and a homeostatic target and (2) plasticity of E->I synapses that is proportional to the difference between the total excitatory input to the I neuron and a homeostatic target. These are sufficient to produce E/I assemblies in a network in which only the excitatory recurrence exhibits tuning at the initial condition. While the full implementation of the plasticity rules, derived from gradient descent on an objective function, would rely on nonlocal weight information, local approximations of the rules still lead to the desired results.

Overall the results make sense and represent a new unsupervised method for generating cell assemblies consisting of both excitatory and inhibitory neurons. My main concerns are that the proposed rule ends up predicting a rather nonstandard form of plasticity for certain synapses, and that the results could be fleshed out more.

1) The main text would benefit from greater exposition of the plasticity rule and the distinction between the full expression and the approximation. While the general idea of backpropagation may be familiar to a good number of readers, here it is being used in a nonstandard way (to implement homeostasis), and this should be described more fully, with a few key equations.

Additionally, the point that, for a recurrent network, the proposed rules are only related to gradient descent under the assumption that the network adiabatically follows the stimulus, seems important enough to state in the main text.

2) The paper has a clear and simple message, but not much exploration of that message or elaboration on the results. Figure 2 and Figure 3 do not convey much information, other than the fact that blocking either form of plasticity fails to produce the desired effects. This seems somewhat obvious -- almost by definition one can't have E/I assemblies if E->I or I->E connections are forced to remain random. I would think this point deserves at most one figure, or maybe even just a few panels.

3) The derived plasticity rule for E->I synapses, which requires modulation of I synapses based on a difference from a target value for the excitatory subcomponent of the input current, does not take a typical form for biologically plausible learning rules (which usually operate on firing rates or voltages, for example). The authors should explore and discuss in more depth this assumption. Is there experimental evidence for it? It seems like it might be a difficult quantity to signal to the synapse in order to guide plasticity. The authors note in the discussion that BCM-type rules fail here -- are there other approaches that would work? What about a more local form of plasticity that involves only the excitatory current local to a dendrite, for example?

4) Does the initial structure in excitatory recurrence play a role, or is it just there to match the data?

Reviewer #2:

In this work, the authors simulated a rate-based recurrent network with 512 excitatory and 64 inhibitory neurons. The authors use this model to investigate which forms of synaptic plasticity are needed to reproduce the stimulus-specific interactions observed between pyramidal neurons and parvalbumin-expressing (PV) interneurons in mouse V1. When there is homeostatic synaptic plasticity from both excitatory to inhibitory and reciprocally from inhibitory to excitatory neurons in the simulated networks, they showed that the emergent E/I assemblies are qualitatively similar to those observed in mouse V1, i.e., stronger synapses for neurons responding to similar stimuli. They also identified that synaptic plasticity must be present in both directions (from pyramidal neurons to PV neurons and vice versa) to produce such E/I assemblies. Furthermore, they identified that these E/I assemblies enable the excitatory population in their simulations to show feature-specific suppression. Therefore, the author claimed that they found evidence that these inhibitory circuits do not provide a "blanket of inhibition", but rather a specific, activity-dependent sculpting of the excitatory response. They also claim that the learning rule they developed in this model shows for the first time how a local approximation rule can instantiate feedback alignment in their network, which is a method for achieving an approximation to a backpropagation-like learning rule in realistic neural networks.

1) The authors claim that their synaptic plastic rule implements a recurrent variant of feedback alignment. Namely, "When we compare the weight updates the approximate rules perform to the updates that would occur using the gradient rule, the weight updates of the local approximations align to those of the gradient rules over learning". They also claim that this is the first time feedback alignment is demonstrated in a recurrent network. It seems that the weight replacement in this synaptic plastic rule is uniquely motivated by E/I balance, but the feedback alignment in [Lillicrap et al., 2016] is much more general. Thus, the precise connections between feedback alignment and this work remains a bit unclear.

It would be good if the following things about this major claim of the manuscript could be expanded and/or clarified:

i) In Figure 1—figure supplement 3 (upper, right vs. left), it is surprising that the Pyr->PV knock-out seems to produce a better alignment in PV->Pyr. Comparing the upper right of Figure 1—figure supplement 3 and the bottom figure of Figure 1G, it seems that the Pyr->PV knock-out performs equally well with a local approximation for the output connections of PV interneurons. Is this a special condition in this model that results in the emergence of the overall feedback alignment?

ii) In the feedback alignment paper [Lillicrap et al., 2016], they introduced a "Random Feedback Weights Support": this uses a random matrix B to replace the transpose of the backpropagation weight matrix. Here, the alignment seems to be based on the intuition that "The excitatory input connections onto the interneurons serve as a proxy for the transpose of the output connections," and "the task of balancing excitation by feedback inhibition favours symmetric connection." It seems synaptic plasticity here is mechanistically different; it is only similar to the feedback alignment [Lillicrap et al., 2016] because both reach a final balanced state. Please clarify how the results here are interpreted as an instantiation of feedback alignment – if it is simply that the end state is similar or if the mechanism is thought to be more deeply connected.

iii) The feedback alignment [Lillicrap et al., 2016] works when the weight matrix has its entries near zero (e^TWBe>0). Are there any analogous conditions for the synaptic plastic rule to succeed?

iv) In the Appendix, the local approximation rule is developed using a 0th-order truncation of Equations 15a and 15b. Is it noted that "If synapses are sufficiently weak.…, this approximation can be substituted into Equation 15a and yields an equation that resembles a backpropagation rule in a feedforward network (E -> I -> E) with one hidden layer -- the interneurons." It would be helpful if the authors can discuss how this learning rule works in a general recurrent network, or if it will work for any network with sufficiently weak synapses.

v) This synaptic plasticity rule seems to be closely related to another local approximation of backpropagation in recurrent neural network: e-prop in (Bellec et al., 2020, https://www.nature.com/articles/s41467-020-17236-y) and broadcast alignment (Nøkland, 2016, Samadi et al., 2017). These previous works do not consider E/I balance in their approximations, but is E/I balance necessary for successful local approximation to these rules?

2) In the Discussion, it reads as if the BCM rule cannot apply to this recurrent network because of the limited number of interneurons in the simulation ("parts of stimulus space are not represented by any interneurons"). Is this a limitation of the size of the model? Would scaling up the simulation change how applicable the BCM learning rule is? It would be helpful if the authors offer a more detailed discussion on why Hebbian forms of plasticity in interneurons fail to produce stimulus specificity.

[Editors’ note: further revisions were suggested prior to acceptance, as described below.]

Thank you for submitting your article "Learning excitatory-inhibitory neuronal assemblies in recurrent networks" for consideration by *eLife*. Your article has been reviewed by two peer reviewers, one of whom is a member of our Board of Reviewing Editors, and the evaluation has been overseen by Richard Ivry as the Senior Editor. The reviewers have opted to remain anonymous.

The reviewers have discussed the reviews with one another and the Reviewing Editor has drafted this decision to help you prepare a revised submission.

Summary:

The manuscript describes conditions under which synaptic plasticity can organize excitatory/inhibitory networks into assemblies in which both neurons have structured connectivity. The work makes predictions about what kinds of plasticity can give rise to this and accounts for recent experimental findings about manipulating inhibitory neurons and the stimulus-dependence of their connectivity.

While recurrent networks in which connections between excitatory (E) neurons are structured into neural "assemblies" are a classic model for the cortex (e.g. the Hopfield model), inhibitory (I) neurons are often assumed to be unstructured in computational models. This simplification is at odds with the observation of E/I connectivity that is organized with respect to stimulus preference and the results of experiments in which activity of I neurons are perturbed. In brain regions spatially organized according to stimulus preference this may be a consequence of local connectivity, but in regions like mouse V1 that lack an obvious topography, it may arise from experience-dependent synaptic plasticity.

The manuscript investigates the situations in which stimulus-specific assemblies can emerge through synaptic plasticity in a recurrent network of E and I (presumed parvalbumin-positive) neurons. The authors combine (1) Hebbian plasticity of I->E synapses that is proportional to the difference between the E neuron's firing rate and a homeostatic target and (2) plasticity of E->I synapses that is proportional to the difference between the total excitatory input to the I neuron and a homeostatic target. These are sufficient to produce E/I assemblies in a network in which only the excitatory recurrence exhibits tuning at the initial condition. While the full implementation of the plasticity rules, derived from gradient descent on an objective function, would rely on nonlocal weight information, local approximations of the rules still lead to the desired results.

The manuscript makes predictions about the results of blocking plasticity of E->I or I->E synapses as well as accounting for the results of experiments in which activating a pyramidal neuron suppresses similarly tuned neurons through recurrent inhibition.

An interesting prediction of the analysis, that is derived through the approximation to the gradient-based weight update, is that synaptic plasticity rules depend on the deviation of recurrent excitatory input current to a neuron from a target value. This quantity is different from those often used in computational models of synaptic plasticity, such as firing rates or voltages. Experiments targeting this quantity and its influence on the results of synaptic plasticity protocols would be an interesting direction for future study.

Essential revisions:

The modifications the authors have made have improved the manuscript. Addressing the following points will further enhance the clarity of the presentation, and will make the work suitable for publication in *eLife*.

1) Could the authors please expand on the negative result about developing selectivity in PV neurons through quenched random connections? Would it be possible to adjust the parameters of the model, such as the weight distribution or number of connections onto PV neurons, so that they would exhibit enough selectivity?

2) Relatedly, the text around Figure 3D says that, "Because of the poor stimulus tuning of the interneurons, output plasticity cannot generate stimulus-specific inhibitory inputs to the Pyr neurons (Figure 3D)." But it looks like there is a definite stimulus-specific inhibitory input that emerges. The authors should clarify what they mean.

3) It would be good if the Abstract could be revised to be accessible to a broader audience of readers.

---

## [Author Response]

[Editors’ note: The authors appealed the original decision. What follows is the authors’ response to the first round of review.]

Reviewer #1:The manuscript investigates the situations in which stimulus-specific assemblies can emerge in a recurrent network of excitatory (E) and inhibitory (I, presumed parvalbumin-positive) neurons. The authors combine (1) Hebbian plasticity of I->E synapses that is proportional to the difference between the E neuron's firing rate and a homeostatic target and (2) plasticity of E->I synapses that is proportional to the difference between the total excitatory input to the I neuron and a homeostatic target. These are sufficient to produce E/I assemblies in a network in which only the excitatory recurrence exhibits tuning at the initial condition. While the full implementation of the plasticity rules, derived from gradient descent on an objective function, would rely on nonlocal weight information, local approximations of the rules still lead to the desired results.Overall the results make sense and represent a new unsupervised method for generating cell assemblies consisting of both excitatory and inhibitory neurons. My main concerns are that the proposed rule ends up predicting a rather nonstandard form of plasticity for certain synapses, and that the results could be fleshed out more.1) The main text would benefit from greater exposition of the plasticity rule and the distinction between the full expression and the approximation. While the general idea of backpropagation may be familiar to a good number of readers, here it is being used in a nonstandard way (to implement homeostasis), and this should be described more fully, with a few key equations.Additionally, the point that, for a recurrent network, the proposed rules are only related to gradient descent under the assumption that the network adiabatically follows the stimulus, seems important enough to state in the main text.

Thanks, that's a good point. We modified the relevant portion of the main text as follows:

“[…] To that end, we derive synaptic plasticity rules for excitatory input and inhibitory output connections of PV interneurons that are homeostatic for the excitatory population (see Materials and methods). A stimulus-specific homeostatic control can be seen as a "trivial" supervised learning task, in which the objective is that all pyramidal neurons should learn to fire at a given target rate ρ0 for all stimuli. Hence, a gradient-based optimisation would effectively require a backpropagation of error [Rumelhart et al., 1985] through time [BPTT; Werbos, 1990].

Because backpropagation rules rely on non-local information that might not be available to the respective synapses, their biological plausibility is currently debated [Lillicrap et al., 2020, Sacramento et al., 2018, Guerguiev et al., 2017, Whittington and Bogacz, 2019, Bellec et al., 2020]. However, a local approximation of the full BPTT update can be obtained under the following assumptions: First, we assume that the sensory input to the network changes on a time scale that is slower than the intrinsic time scales in the network. This eliminates the necessity of backpropagating information through time, albeit still through the synapses in the network. This assumption results in what we call the “gradient-based” rules (Equation 15 in the Appendix), which are spatially non-local. Second, we assume that synaptic interactions in the network are sufficiently weak that higher-order synaptic interactions can be neglected. Third and finally, we assume that over the course of learning, the Pyr→PV connections and the PV→Pyr connections become positively correlated [Znamenskiy et al., 2018], such that we can replace PV->Pyr synapses by the reciprocal Pyr->PV synapse in the Pyr->PV learning rule, without rotating the update too far from the true gradient (see Appendixs)."

We also added the learning rules to the main text.

2) The paper has a clear and simple message, but not much exploration of that message or elaboration on the results. Figure 2 and Figure 3 do not convey much information, other than the fact that blocking either form of plasticity fails to produce the desired effects. This seems somewhat obvious -- almost by definition one can't have E/I assemblies if E->I or I->E connections are forced to remain random. I would think this point deserves at most one figure, or maybe even just a few panels.

We appreciate that the result that both forms of plasticity are necessary may feel somewhat obvious. However, it may not be as obvious as it appears, because the incoming synapses onto INs follow a long-tailed distribution, like many other synapse types. Randomly sampling from such a distribution could in principle generate sufficient stimulus selectivity to render learning in the E->I connections superfluous (see Litwin-Kumar et al., 2017). That’s why we made sure to initialize the E->I weights such that they show a similar variability as in the data. We now comment on this aspect in the Results section:

"Having shown that homeostatic plasticity acting on both input and output synapses of interneurons are *sufficient* to learn E/I assemblies, we now turn to the question of whether both are *necessary*. To this end, we perform "knock-out" experiments, in which we selectively block synaptic plasticity in either of the synapses. The motivation for these experiments is the observation that the incoming PV synapses follow a long-tailed distribution (Znamenskiy et al., 2018). This could provide a sufficient stimulus selectivity in the PV population for PV->Pyr plasticity alone to achieve a satisfactory E/I balance. A similar reasoning holds for static, but long-tailed outgoing PV synapses. This intuition is supported by result of Litwin-Kumar et al., (2017) that in a population of neurons analogous to our interneurons, the dimensionality of responses in that population can be high for static input synapses, when those are log-normally distributed."

Secondly, we tried to write a manuscript for both fellow modelers (how to self-organize an E/I assembly?) and to our experimental colleagues (what conclusions can we draw from the Znamenskiy data?). In electrophysiological studies, the plasticity of incoming and outgoing synapses of INs both have been studied independently. The insight that those two forms of plasticity should act in synergy is something that we wanted to emphasize, because it could be studied in parallel in paired recordings. Hence the two figures. Looks like we got only modelers as reviewers. Along these lines, we added a short paragraph to the Discussion:

“Both Pyr->PV and PV->Pyr plasticity have been studied in slice (for reviews, see, Kullmann et al., 2007, Vogels et al., 2013), but mostly in isolation. The idea that the two forms of plasticity should act in synergy suggests that it may be interesting to study both forms in the same system, e.g., in reciprocally connected Pyr-PV pairs.”

3) The derived plasticity rule for E->I synapses, which requires modulation of I synapses based on a difference from a target value for the excitatory subcomponent of the input current, does not take a typical form for biologically plausible learning rules (which usually operate on firing rates or voltages, for example). The authors should explore and discuss in more depth this assumption. Is there experimental evidence for it? It seems like it might be a difficult quantity to signal to the synapse in order to guide plasticity. The authors note in the discussion that BCM-type rules fail here -- are there other approaches that would work? What about a more local form of plasticity that involves only the excitatory current local to a dendrite, for example?

We agree that the rule we propose for E->I synapses warrants a more extensive discussion regarding its potential biological implementation. We have added the following paragraph to the manuscript:

"A cellular implementation of such a plasticity rule would require the following ingredients: (i) a signal that reflects the cell-wide excitatory current (ii) a mechanism that changes Pyr->PV synapses in response to variations in this signal. For the detection of excitatory inputs, postsynaptic sodium or calcium concentrations are natural candidates. Due to the lack of spines in PV dendrites, both are expected to diffuse more broadly in the dendritic arbor than in spiny neurons [Hu et al., 2014, Kullmann and Lamsa, 2007], and may thus provide a signal for overall dendritic excitatory currents. Depending on how excitatory inputs are distributed on PV interneuron dendrites [Larkum and Nevian, 2008, Jia et al., 2010, Grienberger et al., 2015], the integration of the excitatory currents may not need to be cell-wide -- which could limit the temporal resolution of the plasticity --, but could be local, e.g. to a dendrite, if local excitatory input is a sufficient proxy for the global input. Notably, in PV interneurons, NMDA receptors are enriched in excitatory feedback relative to feedforward connections [LeRoux et al., 2013], suggesting those two sources of excitation are differentially treated on the postsynaptic side.

As for many other excitatory synapses [Sjöström et al., 2008], postsynaptic calcium is likely a key factor also for the plasticity of excitatory input synapses onto interneurons. Blocking NMDA receptors interferes with Hebbian long-term plasticity in some of these synapses [Kullmann and Lamsa, 2007], as does a block of excitatory input [LeRoux et al., 2013]. Furthermore, NMDAR-dependent plasticity in Pyr->PV synapses is expressed postsynaptically and seems to require presynaptic activation [Kullmann and Lamsa, 2007]. In summary, we believe that there are no conceptual issues that would rule out an implementation of the suggested plasticity rule for excitatory inputs onto PV interneurons.”

Concerning other potential types of plasticity, we certainly do not expect that the suggested pair of rules is the only one that will work. The above paragraph is now followed by:

“We also expect that the rules we suggest here are only one set of many that can establish E/I assemblies. Given that the role of the input plasticity in the interneurons is the formation of a stimulus specificity, it is tempting to assume that this could equally well be achieved by classical forms of plasticity like the Bienenstock-Cooper-Munro (BCM) rule [Bienenstock et al., 1982], which is commonly used in models of receptive field formation. However, in our hands, the combination of BCM plasticity in Pyr->PV synapses with homeostatic inhibitory plasticity in the \ItoE synapses showed complex dynamics, an analysis of which is beyond the scope of this article. In particular, this combination of rules often did not converge to a steady state, probably for the following reason. BCM rules tend to […].

We suspect that this instability can also arise for other Hebbian forms of plasticity in interneuron input synapses when they are combined with homeostatic inhibitory plasticity [Vogels et al., 2011] in their output synapses. The underlying reason is that for convergence, the two forms of plasticity need to work synergistically towards the same goal, i.e., the same steady state. For two arbitrary synaptic plasticity rules acting in different sets of synapses, it is likely that they aim for two different overall network configurations. Such competition can easily result in latching dynamics with a continuing turn-over of transiently stable states, in which the form of plasticity that acts more quickly gets to reach its goal transiently, only to be undermined by the other one later [Clopath et al., 2016].”

4) Does the initial structure in excitatory recurrence play a role, or is it just there to match the data?

For the results of Figure 4, the structure of excitatory recurrence is essential, because similarly tuned Pyr neurons should excite each other (absent the E-I assemblies). Without that structure in the Pyr->Pyr connections, the “paradoxical” inhibitory effect we report would not be paradoxical at all. For the results of Figure 1, Figure 2, Figure 3, we primarily included them for consistency, because it's known that this structure is present in V1, and because it's good to see that their presence doesn't change the results. It plays a role only insofar as it permits and reinforces stimulus selectivity in pyramidal neurons. If those synapses were unstructured (and strong), it could disrupt the Pyr selectivity, and there would be nothing to guide the formation of E/I assemblies. We have added the following sentence to the beginning of the Results section:

“[…] Note that the Pyr->Pyr connections only play a decisive role for the results in Figure 4, but are present in all simulations for consistency. […]”

Reviewer #2:In this work, the authors simulated a rate-based recurrent network with 512 excitatory and 64 inhibitory neurons. The authors use this model to investigate which forms of synaptic plasticity are needed to reproduce the stimulus-specific interactions observed between pyramidal neurons and parvalbumin-expressing (PV) interneurons in mouse V1. When there is homeostatic synaptic plasticity from both excitatory to inhibitory and reciprocally from inhibitory to excitatory neurons in the simulated networks, they showed that the emergent E/I assemblies are qualitatively similar to those observed in mouse V1, i.e., stronger synapses for neurons responding to similar stimuli. They also identified that synaptic plasticity must be present in both directions (from pyramidal neurons to PV neurons and vice versa) to produce such E/I assemblies. Furthermore, they identified that these E/I assemblies enable the excitatory population in their simulations to show feature-specific suppression. Therefore, the author claimed that they found evidence that these inhibitory circuits do not provide a "blanket of inhibition", but rather a specific, activity-dependent sculpting of the excitatory response. They also claim that the learning rule they developed in this model shows for the first time how a local approximation rule can instantiate feedback alignment in their network, which is a method for achieving an approximation to a backpropagation-like learning rule in realistic neural networks.

We thank you for your thorough evaluation of the role of feedback alignment (FA) in our model. While we will attempt to address them point-by-point below, we feel that we may have misled this reviewer regarding the focus of the article.

For us, the core novelty of this work lies in elucidating potential mechanisms of experimentally observed E/I neuronal assemblies in mouse V1, and furthermore in proposing plasticity rules that can achieve such E/I assemblies. That they do so via a

mechanism akin to feedback alignment is mentioned relatively briefly in the manuscript, and is merely offered as a mechanistic explanation for how inhibitory currents are ultimately balanced with excitation. We are fully aware of the fact that the suggested rules are by no means a local approximation of the full BPTT problem in RNNs. We rephrased the paper and hope that the emphasis of the paper is now clearer.

1) The authors claim that their synaptic plastic rule implements a recurrent variant of feedback alignment. Namely, "When we compare the weight updates the approximate rules perform to the updates that would occur using the gradient rule, the weight updates of the local approximations align to those of the gradient rules over learning". They also claim that this is the first time feedback alignment is demonstrated in a recurrent network. It seems that the weight replacement in this synaptic plastic rule is uniquely motivated by E/I balance, but the feedback alignment in [Lillicrap et al., 2016] is much more general. Thus, the precise connections between feedback alignment and this work remains a bit unclear.

We had hoped that our claims in the manuscript were phrased sufficiently carefully, and regret that the reviewer was led to believe that our goal was to provide a general solution to biological backprop in recurrent networks. Of course, the problem we are tackling is not the full backprop problem, and we do not expect that the approximation holds for general tasks. It clearly won't, given that it effectively relies on a truncation after two time steps and makes a stationarity assumption. Still, we felt that it would have been a lost opportunity not to discuss the relation to feedback alignment, because any approximation warrants a justification, and for the replacement of I->E weights by E->I weights, feedback alignment readily provides one. We now discuss the assumptions underlying the local approximation more extensively in the main paper (see reply to reviewer 1, comment 1).

We also added a discussion to the section in the Appendix, where the local approximations are derived:

“Overall, the local approximation of the learning rule relies on three assumptions: Slowly varying inputs, weak synaptic weights and alignment of input and output synapses of the interneurons. These assumptions clearly limit the applicability of the learning rules for other learning tasks. In particular, the learning rules will not allow the network to learn temporal sequences.”

It would be good if the following things about this major claim of the manuscript could be expanded and/or clarified:i) In Figure 1—figure supplement 3 (upper, right vs. left), it is surprising that the Pyr->PV knock-out seems to produce a better alignment in PV->Pyr. Comparing the upper right of Figure 1—figure supplement 3 and the bottom figure of Figure 1G, it seems that the Pyr->PV knock-out performs equally well with a local approximation for the output connections of PV interneurons. Is this a special condition in this model that results in the emergence of the overall feedback alignment?

The 0-th order approximation of PV->Pyr plasticity is, by itself, relatively good at following the full gradient for those synapses, because PV->Pyr synapses have virtually unmediated control over Pyr neuron activity. When Py->PV plasticity is also present, we believe that the higher variance in angle to the gradient (for PV->Pyr updates) may be due to perturbations introduced by the Pyr->PV updates. Each update to one weight matrix changes the gradient for the other, but this is ultimately what brings them into alignment with one another.  Because this is a very technical point, we prefer not to discuss this at length in the manuscript. The more important point is conveyed in the two bottom figures, which demonstrate that the gradients on the Pyr->PV synapses only align within 90 degrees when both synapse types are plastic.

ii) In the feedback alignment paper [Lillicrap et al., 2016], they introduced a "Random Feedback Weights Support": this uses a random matrix B to replace the transpose of the backpropagation weight matrix. Here, the alignment seems to be based on the intuition that "The excitatory input connections onto the interneurons serve as a proxy for the transpose of the output connections," and "the task of balancing excitation by feedback inhibition favours symmetric connection." It seems synaptic plasticity here is mechanistically different; it is only similar to the feedback alignment [Lillicrap et al., 2016] because both reach a final balanced state. Please clarify how the results here are to interpreted as an instantiation of feedback alignment – if it is simply that the end state is similar or if the mechanism is thought to be more deeply connected.

We believe that the mechanisms are indeed more deeply connected, as supported by the fact that the gradients align early on during learning. We added an extended discussion to the Appendix:

“In feedback alignment, the matrix that backpropagates the errors is replaced by a random matrix B. Here, we instead use the feedforward weights in the layer below. Similar to the extension to feedback alignment of Akrout et al., [2019], those weights are themselves plastic. However, we believe that the underlying mechanism of feedback alignment still holds. The representation in the hidden layer (the interneurons) changes as if the weights to the output layer (the Pyr neurons) were equal to the weights they are replaced with (here, the input weights to the PV neurons). To exploit this representation, the weights to the output layer then align to the replacement weights, justifying the replacement post-hoc (Figure 1G).”

iii) The feedback alignment [Lillicrap et al., 2016] works when the weight matrix has its entries near zero (e^TWBe>0). Are there any analogous conditions for the synaptic plastic rule to succeed?

Yes, the condition is very similar. We have added a corresponding discussion to the Appendix:

“Note that the condition for feedback alignment to provide an update in the appropriate direction (e^T^ B^T^ W e>0, where e denotes the error, W the weights in the second layer, and B the random feedback matrix) reduces to the condition that Wei W_ie_ is positive definite (assuming the errors are full rank).

One way of assuring this is a sufficiently positive diagonal of this matrix product, i.e., a sufficiently high correlation between the incoming and outgoing synapses of the interneurons. A positive correlation of these weights is one of the observations of Znamenskiy et al., 2018 and also a result of learning in our model.

While such a positive correlation is not necessarily present for all learning tasks or network models, we speculate that it will be for the task of learning an E/I balance in a Dalean network.”

iv) In the Supplementary material, the local approximation rule is developed using a 0th-order truncation of Equations 15a and 15b. Is it noted that "If synapses are sufficiently weak.…, this approximation can be substituted into Equation 15a and yields an equation that resembles a backpropagation rule in a feedforward network (E -> I -> E) with one hidden layer -- the interneurons." It would be helpful if the authors can discuss how this learning rule works in a general recurrent network, or if it will work for any network with sufficiently weak synapses.

We now discuss the assumptions and their consequences more extensively, see reply to reviewer 1, comment 1.

v) This synaptic plasticity rule seems to be closely related to another local approximation of backpropagation in recurrent neural network: e-prop in (Bellec et al., 2020, https://www.nature.com/articles/s41467-020-17236-y) and broadcast alignment (Nøkland, 2016, Samadi et al., 2017). These previous works do not consider E/I balance in their approximations, but is E/I balance necessary for successful local approximation to these rules?

We are not sure if we fully understand the comment. We do not expect that E/I balance is necessary for other biologically plausible approximations of BPTT. We merely suggest that for the task of learning E/I balance, the presented local approximation is valid.

2) In the Discussion, it reads as if the BCM rule cannot apply to this recurrent network because of the limited number of interneurons in the simulation ("parts of stimulus space are not represented by any interneurons"). Is this a limitation of the size of the model? Would scaling up the simulation change how applicable the BCM learning rule is? It would be helpful if the authors offer a more detailed discussion on why Hebbian forms of plasticity in interneurons fail to produce stimulus specificity.

Increasing the size of the model would help only if it would increase the redundancy in the Pyr population response. Otherwise, the problem can only be solved by changing the E to I ratio.

We feel that an exhaustive discussion of the dynamics of BCM in our network is beyond the scope of the paper, particularly because BCM comes in a broad variety (weight normalisation, weight limits, exact form of the sliding threshold?) and the exact behavior depends on various parameter choices. Similarly, we preferred to limit the discussion of other Hebbian rules, because it would be somewhat arbitrary which rules to discuss. Instead we added the following more abstract arguments to the Discussion section:

“We expect that the rules we suggest here are only one set of many that can establish E/I assemblies. Given that the role of the input plasticity in the interneurons is the formation of a stimulus specificity, it is tempting to assume that this could equally well be achieved by classical forms of plasticity like the Bienenstock-Cooper-Munro (BCM) rule {Bienenstock82}, which is commonly used in models of receptive field formation. However, in our hands, the combination of BCM plasticity in Pyr->PV synapses with homeostatic inhibitory plasticity in the PV->Pyr synapses showed complex dynamics, an analysis of which is beyond the scope of this article. In particular, this combination of rules often did not converge to a steady state, probably for the following reason. […]

We suspect that this instability can also arise for other Hebbian forms of plasticity in interneuron input synapses when they are combined with homeostatic inhibitory plasticity (Vogels et al., 2011) in their output synapses. The underlying reason is that for convergence, the two forms of plasticity need to work synergistically towards the same goal, i.e., the same steady state. For two arbitrary synaptic plasticity rules acting in different sets of synapses, it is likely that they aim for two different overall network configurations. Such competition can easily result in dynamics with a continuing turn-over of transiently stable states, in which the form of plasticity that acts more quickly gets to reach its goal transiently, only to be undermined by the other one later.”

[Editors’ note: what follows is the authors’ response to the second round of review.]

Essential revisions:The modifications the authors have made have improved the manuscript. Addressing the following points will further enhance the clarity of the presentation, and will make the work suitable for publication in eLife.1) Could the authors please expand on the negative result about developing selectivity in PV neurons through quenched random connections? Would it be possible to adjust the parameters of the model, such as the weight distribution or number of connections onto PV neurons, so that they would exhibit enough selectivity?

The reviewers raised an interesting question, which motivated us to investigate if sampling PV input weights from a distribution with an even longer tail may affect these results. In brief, the broader distribution can indeed increase the selectivity in the PV interneurons, but it is not sufficient to account for the correlation structure in the synaptic weights observed by Znamenskiy et al. The underlying reason is that the selectivity of the interneurons is then controlled by a very small number of large excitatory input weights, which are unlikely to be sampled in an experiment. We have added a new figure supplement and corresponding text to the manuscript (see below).

We would like to emphasize that our original choice of the weight distribution is motivated by observations in the mouse cortex (Song et al., 2005, Loewenstein et al., 2011), covering about two orders of magnitude. Similarly, the connection probability of 60% we used is in line mouse V1 connectivity (Znamenskiy et al., 2018). We do not exclude the possibility that the parameters in our model could in principle be fine-tuned to produce a measurable correlation between PV input and output synapses. However the underlying network model would probably not be consistent with observations from mouse V1.

“[…] A particularly strong input connection will cause the postsynaptic interneuron to prefer similar stimuli to the presynaptic Pyr. Because of the resulting correlated activity, the Hebbian nature of the output plasticity potentiates inhibitory weights for such cell pairs that are reciprocally connected. This tendency of strong input synapses to generate a strong corresponding output synapse is reflected in a positive correlation between output synapses and response similarity (Figure 3E, Figure 3—figure supplement 1A), despite the fact that input synapses remain random.

This effect further increases when input synapses are drawn from a distribution with an even heavier tail, beyond what is observed in mouse V1 [Znamenskiy et al., 2018] (Figure 3—figure supplement 2A). In this case, the stimulus tuning of the interneurons is dominated by a small number of very large synapses. The resulting higher selectivity of the interneurons (Figure 3—figure supplement 2B) allows a better co-tuning of excitation and inhibition in Pyr neurons (Figure 3—figure supplement 2C), in line with theoretical arguments for sparse connectivity [Litwin-Kumar et al., 2017]. However, the dominance of a small number of large synapses also makes it unlikely that those synapses are observed in an experiment in which a finite number of synapses is sampled. As a result, a heavier tail does not yield the correlation of reciprocal in- and output synapses observed by Znamenskiy et al., [2018] (Figure 3—figure supplement 2D,E), although it increases the probability of observing correlations between input synapses and response similarity when weak synapses are discarded. See Appendix for a more extensive discussion.”

2) Relatedly, the text around Figure 3D says that, "Because of the poor stimulus tuning of the interneurons, output plasticity cannot generate stimulus-specific inhibitory inputs to the Pyr neurons (Figure 3D)." But it looks like there is a definite stimulus-specific inhibitory input that emerges. The authors should clarify what they mean.

Thank you for pointing this out. We agree that this was not clear in the text and have extended the paragraph:

“Because of the poor stimulus tuning of the interneurons, output plasticity cannot generate stimulus-specific inhibitory inputs to the Pyr neurons (Figure 3D). Instead, they essentially receive a tonic, unspecific background inhibition that is weakly modulated by the stimulus (Figure 1—figure supplement 2B). This weak modulation is indeed correlated with the excitatory inputs, but the overall similarity between excitation and inhibition remains low (Figure 1—figure supplement 2C).”

3) It would be good if the Abstract could be revised to be accessible to a broader audience of readers.

We appreciate the feedback on the current Abstract and agree that particularly the first few sentences needed to be refined. We have revised the Abstract to convey the problem setting and our main findings to a broader audience. You can find the revised Abstract below:

“In contrast to sensory circuits with feature topography, neurons in the mouse primary visual cortex (V1) are not arranged according to the stimulus features they respond to. Yet, synapses between excitatory pyramidal neurons and inhibitory parvalbumin-expressing (PV) interneurons tend to be stronger for neurons that respond to similar stimulus features. The presence of such excitatory-inhibitory (E/I) neuronal assemblies indicates a stimulus-specific form of feedback inhibition despite the absence of feature topography. Here, we show that activity-dependent synaptic plasticity on input and output synapses of PV interneurons generates a circuit structure that is consistent with mouse V1. Using a computational model, we show that both forms of plasticity must act in synergy to form the observed E/I assemblies. Once established, these assemblies produce a stimulus-specific competition between pyramidal neurons. Our model suggests that activity-dependent plasticity can refine inhibitory circuits to actively shape cortical computations.”